# ALSO: Adversarial Online Strategy Optimization for Social Agents

**Xiang Li** [1]   **Liping Yi** [1]   **Mingze Kong** [2]   **Min Zhang** [3]   **Zhongxiang Dai** [2]   **QingHua Hu** [1]

## Abstract

Social simulation provides a compelling testbed for studying social intelligence, where agents interact through multi-turn dialogues under evolving contexts and strategically adapting opponents. Such environments are inherently non-stationary, requiring agents to dynamically adjust their strategies over time. However, most Large Language Model (LLM) based social agents rely on static personas, while existing approaches for enhancing social intelligence, such as offline reinforcement learning or external planners, are ill-suited to these settings, typically assuming stationarity and incurring substantial training overhead. To bridge this gap, we propose **ALSO** (**A**dversarial on**L**ine **S**trategy **O**ptimization), the first framework for online strategy optimization in multi-agent social simulation. ALSO advances social adaptation through two key contributions. (1) ALSO formulates multi-turn interaction as an adversarial bandit problem, where combinations of static personas and dynamic strategy instructions are treated as arms, providing a principled solution to non-stationarity without relying on environmental stability assumptions. (2) To predict rewards and generalize sparse feedback in multi-turn dialogues, ALSO introduces a lightweight neural surrogate to predict rewards from interaction histories, enabling sample-efficient exploration and continuous online adaptation. Experiments on the Sotopia benchmark demonstrate that ALSO consistently outperforms static baselines and existing optimization methods in dynamic environments, validating the effectiveness of adversarial online strategy optimization for building robust social agents.

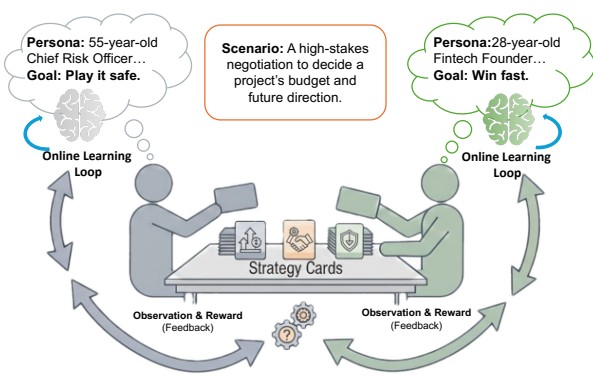

*Figure 1.* Online social interaction between agents with *personas* and *adaptive strategies*, where feedback multi-turn dialogue drives continuous strategy optimization under evolving behaviors.

## 1. Introduction

Modeling social intelligence (Mathur et al., 2024) is a central pursuit in Artificial Intelligence research. The advent of Large Language Models (LLMs) has substantially advanced this field by endowing agents with human-like communication (Spitale et al., 2023) and planning capabilities (Wu et al., 2024a; Park et al., 2023). This progress has positioned LLM-based social simulation as a powerful framework for studying emergent social behaviors in large-scale and goal-oriented interactions (Epstein, 2012; Wang et al., 2024a). In such simulations, agent behavior is typically governed by a *persona*, a formalized profile encapsulating personality traits, occupations, and background stories (Reiss, 2023; Salinas & Morstatter, 2024; Bisbee et al., 2024).

While static personas provide foundational identity, they alone are insufficient for eliciting adaptive social intelligence. It is important to distinguish between *persona* and *strategy*: a persona defines *who* an agent is, whereas a strategy specifies *how* the agent acts to navigate interactions and achieve goals. Empirical studies demonstrate that relying solely on static personas often yields stereotypical and homogeneous behaviors, failing to capture the diversity required for robust social simulation (Taubenfeld et al., 2024; Hwang et al., 2025; Zeng et al., 2025; Venkit et al., 2026). This homogeneity is further reinforced by the "alignment tax" of Reinforcement Learning from Human Feedback (RLHF), which suppresses behavioral variance in fa-

---

[1]School of Artificial Intelligence, Tianjin University, Tianjin, China [2]The Chinese University of Hong Kong, Shenzhen, China [3]East China Normal University, Shanghai, China. Correspondence to: Zhongxiang Dai <daizhongxiang@cuhk.edu.cn>.

*Proceedings of the 43rd International Conference on Machine Learning*, Seoul, South Korea. PMLR 306, 2026. Copyright 2026 by the author(s).

vor of safety (Kirk et al.). Without evolving strategies to complement identity, agents struggle to adapt to dynamic opponents, resulting in rigid and suboptimal interaction patterns (Li et al., 2023; Wang et al., 2024b; Zeng et al., 2025).

To enhance social adaptability, recent work has explored automated prompt optimization (APO) for refining agent instructions (Zhou et al., 2022a; Guo et al., 2024; Lin et al., 2024b; Opsahl-Ong et al., 2024a), which can be interpreted through multi-armed bandit formulations. However, both offline approaches and online variants (Yang et al., 2023; Lin et al., 2024a; Wu et al., 2024b) fundamentally rely on stationarity assumptions, where each prompt induces a stable reward distribution evaluated on fixed validation sets. Such assumptions break down in social simulation, as feedback from multi-turn interactions with strategically adapting counterparts whose behaviors co-adapt with the agent's strategy choices, inducing persistent reward shifts and strong temporal coupling. This dynamic feedback loop renders standard stochastic bandit models inadequate for social strategy instruction optimization.

To bridge this gap, we propose **ALSO** (**A**dversarial On**l**ine **S**trategy **O**ptimization), an *online* framework that *casts social strategy adaptation as an adversarial multi-armed bandit problem* to enable principled optimization under non-stationarity, as shown in Figure 1. Rather than assuming stationary rewards, ALSO explicitly models the co-evolving and strategically adaptive nature of social interactions with two designs. (1) In ALSO, each arm corresponds to a strategy instruction sampled from a generated strategy set (*e.g.*, cooperation, competition, deception, rational bargaining). At each interaction round, the selected strategy is combined with the agent's persona to form the final prompt context, ensuring that dynamic adaptation remains grounded in consistent identity. Based on this adversarial bandit formulation, ALSO instantiates a robust online optimization procedure inspired by EXP3 (Lattimore & Szepesvári, 2020) with smoothing to hedge against shifting opponent behaviors. (2) Standard bandit methods treat arms independently and fail to exploit semantic relationships among strategy instructions. To address this limitation, ALSO introduces a lightweight neural surrogate model that leverages interaction histories to predict rewards and generalize sparse feedback across semantically related strategies. This enables sample-efficient online optimization under sparse multi-turn feedback.

Overall, ALSO forms a closed-loop online system that iteratively selects strategies, interacts under persona-conditioned prompts, and updates both the bandit policy and surrogate model from feedback. We evaluate ALSO on Sotopia, a comprehensive LLM-based social simulation benchmark spanning seven dimensions of social intelligence. Across diverse settings, ALSO consistently outperforms static persona agents and existing optimization baselines, achieving a +16.60% overall improvement and +83.79% substantial gains on relationship outcomes.

**Contributions.** We make the following contributions:

- We introduce the first online strategy learning framework for LLM-based multi-agent social simulation, enabling dynamic adaptation beyond static persona-driven behavior in evolving environments.

- We formulate social strategy optimization as an adversarial bandit problem with surrogate reward modeling, providing a principled solution to non-stationary and strategically adaptive interactions.

- We conduct extensive evaluations on the Sotopia benchmark, demonstrating consistent improvements over static agents and existing optimization baselines across diverse social settings.

## 2. Related work

### 2.1. Social Intelligence

Social intelligence, distinct from abstract and mechanical intelligence, refers to the ability to manage interpersonal relations and social contexts (Thorndike, 1920; Strang, 1930; Thorndike & Stein, 1937). In computational settings, Artificial Social Intelligence emphasizes modeling and responding to the mental and behavioral dynamics of interacting partners, including both humans and artificial agents (Sap et al., 2022; Gweon et al., 2023; Mathur et al., 2024). Recent advances in Large Language Models (LLMs) provide a strong foundation for building socially capable agents (Hoppler et al., 2022; Lee et al., 2024; Anthis et al., 2025).

Early social evaluation frameworks (*e.g.*, Social IQa (Sap et al., 2019) and SocialBench (Chen et al., 2024)) focused on static multiple-choice settings, which fail to capture the dynamic and non-stationary nature of social interaction. This limitation motivated dynamic simulation-based benchmarks, including SOTOPIA (Zhou et al., 2024) and AgentSense (Mou et al., 2025), which assess agents in open-ended multi-turn environments with continuously evolving goals and social relations.

Existing methods for enhancing social intelligence generally follow two paradigms. The first category focuses on *offline optimization*. Sotopia-π (Wang et al., 2024b) improves performance through data-centric refinement, while Sotopia-RL (Yu et al., 2025) and SDPO (Kong et al., 2025a) address credit assignment and preference optimization in multi-turn dialogues. Adaptive Mode Learning (AML) (Wang et al., 2025a) further promotes diverse social reasoning patterns.

The second category augments inference through offline-trained external planners. Methods such as Sotopia-

$\Omega$ (Zhang et al., 2025), DAT (Li et al., 2024), and EPO (Liu et al., 2025) learn auxiliary planning models from generated data to provide high-level strategic guidance at test time. While these approaches highlight the importance of strategies in social interaction, they embed strategic behaviors either within model parameters or fixed planners.

As a result, introducing new strategies or adapting to evolving social dynamics typically requires data recollection and retraining. This limitation motivates our ALSO, which enables dynamic and sample-efficient strategy adaptation through online optimization without offline retraining.

## 2.2. Prompt Optimization

Prompt Optimization (PO) provides an efficient mechanism for adapting agent behavior without parameter fine-tuning by treating strategies as optimizable instructions (Wang et al., 2025b). Early PO methods primarily fall into two categories: *LLM-as-Optimizer* approaches such as APE (Zhou et al., 2022b), OPRO (Yang et al., 2023), and Instinct (Lin et al., 2024c), which iteratively generate and refine prompts, and evolutionary methods including EvoPrompt (Guo et al., 2024) and PromptBreeder (Fernando et al., 2024), which explore instruction spaces via mutation and selection.

Despite their effectiveness in static tasks, most PO methods rely on offline oracles, such as fixed validation sets, to evaluate and rank candidate strategies (Opsahl-Ong et al., 2024b; Kong et al., 2025b). This offline paradigm assumes stationary reward distributions and fails to capture the coupled, evolving dynamics of social interaction, where optimal strategies shift in response to adaptive opponents. Consequently, existing PO frameworks are ill-suited for online social simulation, motivating the need for adversarial and online strategy optimization as pursued in ALSO.

## 3. Problem Formulation

We consider a multi-agent social simulation environment and formulate online social strategy learning as sequential strategy selection under non-stationary interactions.

### 3.1. Multi-Agent Social Simulation Environment

We consider a general multi-agent social simulation framework in which LLMs act as interactive agents. The agent set is denoted by $\mathcal{N} = \{1, 2, \ldots, N\}$, where each agent $i \in \mathcal{N}$ is characterized by a persona $b_i \in \mathcal{B}$ and a private social goal $g_i \in \mathcal{G}$. The scenario $\mathcal{S}$ specifies the social context, including environmental settings and interaction constraints.

The simulation proceeds in discrete dialogue rounds indexed by $l = 1, \ldots, L$. At round $l$, the active agent $i$ forms an observation $o_l^i$ by conditioning on the scenario, interaction

history $\mathcal{H}_{l-1}$, its persona, and its goal:

$$o_l^i = \text{Prompt}(\mathcal{S}, \mathcal{H}_{l-1}, b_i, g_i). \tag{1}$$

The agent then samples an action $a_l^i$ based on $o_l^i$ from the LLM:

$$a_l^i \sim \text{LLM}(o_l^i). \tag{2}$$

The environment updates the state and augments the history as $\mathcal{H}_l = \mathcal{H}_{l-1} \cup \{a_l^i\}$.

After $L$ dialogue rounds, an LLM-based evaluator assesses agent performance along $M$ social dimensions:

$$\{d_m^{(i)}\}_{m=1}^M = \text{LLM}_{\text{eval}}(\mathcal{S}, \mathcal{H}^{(L)}, b_i, g_i), \tag{3}$$

which are aggregated into a scalar reward:

$$r_i = \frac{1}{M} \sum_{m=1}^M \phi_m(d_m^{(i)}). \tag{4}$$

However, a key different of social simulation from this setting is its inherent *non-stationary* arising from strategically adaptive agents. Let $\mathbf{A}_l = (a_l^{(1)}, \ldots, a_l^{(N)})$ denote the joint action set at step $l$, and $\mathbf{a}_l^{-i}$ the actions of all agents except agent $i$. Let $R(s_l, a_l^{(i)}, \mathbf{a}_l^{-i})$ denote the instantaneous turn-level reward function under state $s_l$ and joint actions set. The expected step/turn level reward for agent $i$ is given by

$$\mathbb{E}[r_l^{(i)} \mid s_l, a_l^{(i)}] = \sum_{\mathbf{a}_l^{-i} \in A^{-i}} R(s_l, a_l^{(i)}, \mathbf{a}_l^{-i}) \prod_{j \neq i} \pi_j(a_j \mid s_l), \tag{5}$$

where $\pi_j$ denotes the evolving policy of agent $j$.

In practice, agents only observe the realized $r_i$, while the underlying expected reward $\mathbb{E}[r_l^{(i)} \mid s_l, a_l^{(i)}]$ remains implicit and shifts continuously as opponent policies $\pi_j$ evolve. This creates a fundamental learning challenge:

$$r_i = \mathbb{E}[r_l^{(i)} \mid s_l, a_l^{(i)}] + \epsilon_t, \quad \text{where } \epsilon_t \text{ is non-stationary}, \tag{6}$$

where $\epsilon_t$ captures both stochastic noise from LLM generation and distributional shift induced by co-evolving opponent policies. This motivates casting the strategy learning problem as adversarial online optimization (Section 3.2), where no stationarity of the reward signal is assumed.

### 3.2. Online Social Strategy Learning Problem

We model social strategy adaptation by discretizing the strategy space into a finite set of $K$ strategic instructions (*e.g.*, cooperation and competition), each corresponding to a bandit arm. Each instruction encodes high-level behavioral

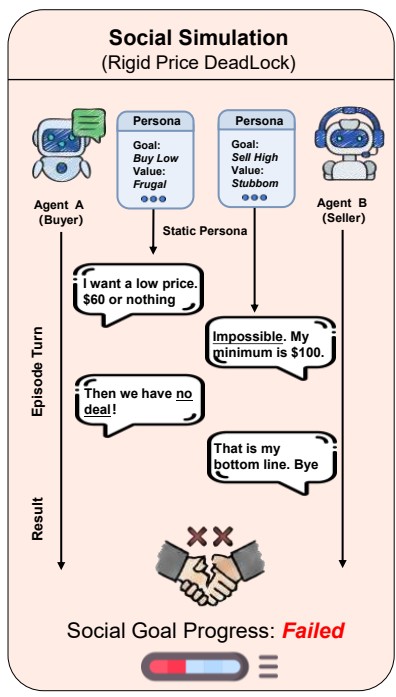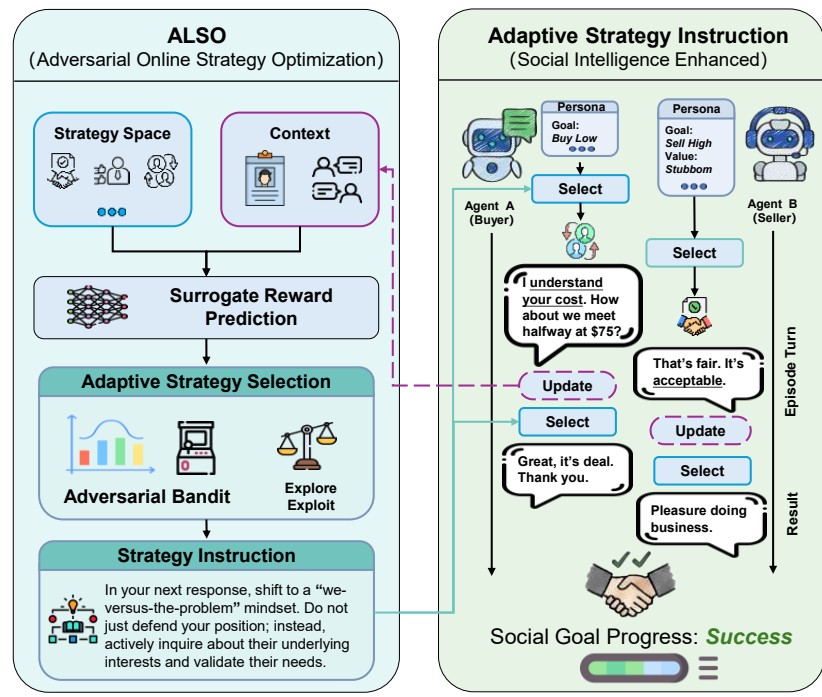

*Figure 2.* Overview of `ALSO` for adaptive social strategy learning in LLM-based multi-agent social simulation. Static persona-driven agents exhibit rigid interactions and fail to achieve social goals (left), while `ALSO` leverages adversarial online strategy selection with surrogate reward modeling to dynamically adapt strategies and enable successful social outcomes (center–right).

principles over goal orientation and interaction style, forming an interpretable strategy space for online learning. In this formulation, the agent's action at each turn is to *select an arm*, *i.e.*, choose a strategy instruction to guide its behavior throughout the social interaction. This directly casts online strategy selection under evolving social dynamics as an adversarial multi-armed bandit problem.

**Objective.** Over rounds $t = 1, \ldots, T$ (each corresponding to a complete $L$-turn social simulation episode), agent $i$ selects a strategy arm $k_t \in [K]$ and receives an trun-level reward $r_{k_t}^{(t)} \in [0, 1]$ from the evaluator (Eq. (4)). A natural *target objective* is to minimize the cumulative pseudo-regret

$$\bar{R}_T = \mathbb{E}\left[ \max_{k \in [K]} \sum_{t=1}^{T} r_k^{(t)} - \sum_{t=1}^{T} r_{k_t}^{(t)} \right], \qquad (7)$$

which measures performance relative to the best fixed strategy in hindsight. In our setting, however, the reward process is induced by co-evolving multi-agent interactions and an LLM evaluator; thus, `ALSO` adopts adversarial online learning as a *design rationale* for robustness rather than claiming a formal regret guarantee.

## 4. Methodology

This section presents `ALSO`, our adversarial online approach to strategy optimization in LLM-based multi-agent social simulations, as displayed in Figure 2.

### 4.1. Problem Setting

We formalize dynamic strategy instruction optimization in two-agent ($N = 2$) LLM-based social simulations. Each agent $i \in \{1, 2\}$ augments its original persona $b_i^0$ with one of $K$ predefined social strategies $\Sigma = \{\sigma_1, \ldots, \sigma_K\}$ (details in Appendix B.2).

In the original framework, the agent's action is generated using a fixed persona:

$$a_t^{(i)} \sim \text{LLM}\big(\mathcal{S}, \mathcal{H}_{t-1}, b_i^0, g_i\big). \qquad (8)$$

In `ALSO`, we dynamically augment the persona at each turn $t$ by appending a selected strategy instruction $\sigma_{k_t}^{(i)}$ from the strategy space $\Sigma$. This creates an enhanced persona

$$b_i^{(t)} = b_i^0 \oplus \sigma_{k_t}^{(i)}, \qquad (9)$$

where $\oplus$ denotes textual concatenation. The resulting persona $b_i^{(t)}$ incorporates both the agent's original identity (demographics, personality, values, etc.) and the high-level

behavioral guidance provided by the strategy (e.g., collaborative problem-solving, firm bargaining, or strategic withholding). The LLM then generates the next action conditioned on this augmented context:

$$a_t^{(i)} \sim \text{LLM}\big(\mathcal{S}, \mathcal{H}_{t-1}, b_i^{(t)}, g_i\big). \qquad (10)$$

This augmentation allows the LLM to adapt its responses based on the chosen strategy (e.g., collaboration or information withholding) without retraining the model.

**Non-stationarity.** In social simulation, the reward process is inherently non-stationary: the counterpart agent adapts to the ego agent's behavior, and the dialogue state evolves over turns. Therefore, we do *not* assume rewards are i.i.d. or drawn from a fixed distribution across optimization iterations; instead, we adopt an adversarial online learning perspective as a *design rationale* for robust strategy selection under distribution shift.

**Feedback.** After each round, we use an LLM evaluator to provide per-turn rewards $r_t^{(i)} = \rho(s_t, a_t^{(i)})$ over $M$ dimensions, which are normalized by $\phi_m$ (Table 4), following prior LLM-evaluator-based social simulation work such as Sotopia-$\Omega$ (Zhang et al., 2025) and AML (Wang et al., 2025b). We use the resulting scalar reward (after aggregation/normalization) as the bandit feedback in Algorithm 1.

## 4.2. ALSO: Adversarial Online Strategy Optimization

To address strategy optimization in a large, discrete space under non-stationary social dynamics, we propose ALSO. ALSO follows an adversarial online learning design: it makes no stationarity assumptions about rewards, uses randomized selection to remain robust to shifting partner behaviors, and incorporates a recency mechanism to track drift. Concretely, ALSO combines an exponential-weights selector with a lightweight neural surrogate that generalizes sparse feedback across strategies. The LLM policy remains frozen; only the surrogate network is trained online.

The core logic of ALSO is organized into the following phases (including a preprocessing step): We denote by $a_t$ the focal agent's utterance and by $o_t$ the counterpart response; $\mathcal{H}^{(t)}$ is the dialogue history up to turn $t$.

**Arm Space (Alg. 1, Line 1–2).** Before online interaction begins, we precompute an embedding for each *augmented persona* obtained by appending a candidate strategy to the base persona for computer efficiency. Concretely, for each $k \in [K]$, we form $b^{(k)} = b^0 \oplus \sigma_k$ and compute $\mathbf{b}_k = g(b^{(k)})$. These embeddings are fixed across turns and serve as the strategy-specific representation used by the surrogate.

**Context Encoding & Prediction (Alg. 1, Lines 5–6).** At the beginning of each interaction turn $t$, the optimal strategy depends on the dialogue state. We use the frozen embedding

**Algorithm 1** Adversarial Online Strategy Optimization

---

**Require:** Strategy space $\Sigma = \{\sigma_1, \ldots, \sigma_K\}$ (candidate strategy instructions), base persona $b^0$ (fixed persona text), frozen embedding model $g(\cdot)$, trainable value network $f_\theta$, learning rate $\eta > 0$, decay factor $\lambda \in (0, 1]$, batch size $B$

**Ensure:** Sequence of selected strategies $\{\sigma_{k_t}\}_{t=1}^T$ (the strategy chosen at each turn $t$)

1: **Precompute augmented-persona embeddings:**
2:    For all $k$, form $b^{(k)} \leftarrow b^0 \oplus \sigma_k$ and compute $\mathbf{b}_k \leftarrow g(b^{(k)})$
3: **Initialize:** $S_k^{(0)} \leftarrow 0$ for all $k$; replay buffer $\mathcal{D} \leftarrow \emptyset$; history $\mathcal{H}^{(0)} \leftarrow \emptyset$
4: **for** turn $t = 1$ to $T$ **do**
5:    **Context encoding:** $\mathbf{c}^{(t)} \leftarrow g(\mathcal{H}^{(t-1)})$
6:    **Value prediction:** compute $(\mathbf{x}_k^{(t)}, \hat{v}_k^{(t)})$ for all $k$ via Eq. 11
7:    **Strategy selection:** compute $\pi^{(t)}$ and sample $k_t$ via Eq. 12
8:    **Interaction & feedback:** form augmented persona $b^{(t)} \leftarrow b^0 \oplus \sigma_{k_t}$ and execute $b^{(t)}$ to generate focal-agent action $a_t$; observe counterpart response $o_t$ and reward $r_t$; update $\mathcal{H}^{(t)} \leftarrow \mathcal{H}^{(t-1)} \cup \{a_t, o_t\}$
9:    **Surrogate update:** add $(\mathbf{x}_{k_t}^{(t)}, r_t)$ to $\mathcal{D}$; sample a minibatch of size $B$ from $\mathcal{D}$ and update $f_\theta$ via MSE
10:    **Score smoothing:** update $S_k^{(t)}$ for all $k$ via Eq. 13
11: **end for**

---

model $g(\cdot)$ to encode the dialogue history $\mathcal{H}^{(t-1)}$ into a context vector $\mathbf{c}^{(t)}$. For each candidate strategy $\sigma_k \in \Sigma$, we concatenate the precomputed augmented-persona embedding $\mathbf{b}_k$ with $\mathbf{c}^{(t)}$ to form $\mathbf{x}_k^{(t)}$. A trainable value network $f_\theta(\cdot)$ predicts the expected reward $\hat{v}_k^{(t)}$ for all arms:

$$\mathbf{x}_k^{(t)} = [\mathbf{b}_k; \mathbf{c}^{(t)}], \qquad \hat{v}_k^{(t)} = f_\theta(\mathbf{x}_k^{(t)}). \qquad (11)$$

This provides a data-efficient inductive bias in early online learning, where only a small number of interactions are available.

**Strategy Selection (Alg. 1, Lines 7).** We maintain a cumulative score $S_k$ for each arm and sample strategies from an exponential-weights distribution:

$$\pi_k^{(t)} \propto \exp\big(\eta S_k^{(t-1)}\big), \qquad k_t \sim \text{Categorical}(\pi^{(t)}). \quad (12)$$

Randomized selection is essential in the adversarial setting, where a greedy policy can be exploited or can overfit to transient dynamics.

**Interaction & Surrogate Update (Alg. 1, Lines 8–9).** After sampling $\sigma_{k_t}$ and observing the per-turn reward $r_t$, we store $(\mathbf{x}_{k_t}^{(t)}, r_t)$ in a replay buffer $\mathcal{D}$ and update $f_\theta$ by minimizing an MSE loss. The surrogate improves sample effi-

ciency by transferring supervision to semantically related strategies.

**Score estimation.** In classical EXP3, the exponential-weights sampling distribution is constructed from per-arm cumulative rewards. In our online social simulation setting, however, we only observe feedback for the selected strategy at each turn, and many candidate strategies (or their paraphrased variants) may never be played. This makes direct per-arm reward accumulation highly sample-inefficient, especially when the effective arm space is large. Motivated by prior prompt optimization work, we therefore use a lightweight neural surrogate $f_\theta$ over pretrained embeddings to estimate scores for *all* arms from the current dialogue context. This provides dense score estimates to construct the exponential-weights distribution, and propagates sparse feedback across semantically related strategies while keeping the base LLM frozen.

**Score Smoothing (Alg. 1, Lines 10).** To explicitly track non-stationarity, we apply an exponential decay factor $\lambda \in (0, 1]$ (set to $\lambda = 0.9$ in all experiments) so that recent evidence dominates historical estimates. This keeps $S_k^{(t)}$ responsive to partner shifts while preventing outdated interactions from dominating the strategy distribution:

$$S_k^{(t)} = \lambda S_k^{(t-1)} + \hat{v}_k^{(t)}. \tag{13}$$

## 5. Experiments

This section evaluates ALSO on LLM-based social simulation benchmarks to assess its effectiveness for online strategy adaptation under non-stationary interactions. The codes of ALSO are available at https://github.com/Babylonehy/ALSO

### 5.1. Experimental Setting

**Benchmarks.** We evaluate ALSO on Sotopia (Zhou et al., 2024) and its challenging subset Sotopia-Hard. Sotopia contains 90 two-agent social scenarios spanning negotiation, collaboration, and competition, while Sotopia-Hard includes 14 scenarios emphasizing complex and conflicting social dynamics. We extend the original implementation to support dynamic strategy injection. Our strategy instruction pool consists of 12 predefined strategies (Appendix B.2), shared across all optimization-based methods for fair comparison.

**Online Interaction Protocol.** Each episode instantiates a single scenario with fixed personas and goals and runs up to 20 dialogue turns. At each turn, agents select a strategy instruction from $\Sigma$ and append it to their base persona before generating responses. Unless otherwise specified, we adopt a bilateral online setting where each agent maintains an independent optimizer and updates it solely from its own interaction feedback.

| METHOD | AGENT | EVALUATOR | OPTIMIZER |
|---|---|---|---|
| Vanilla / INSTINCT / Ours (ALSO) | $2T$ | $T$ | $0$ |
| OPRO (every 5 turns) | $2T$ | $T$ | $\lceil T/5 \rceil$ |
| EvoPrompt (pop.=5, every 5 turns) | $2T$ | $T$ | $5 \cdot \lceil T/5 \rceil$ |

*Table 1.* Per-episode LLM call budget accounting (two-agent episode with horizon $T$ turns). "Agent" counts calls to the dialogue LLMs that generate actions; "Evaluator" counts calls to the per-turn reward model; "Optimizer" counts extra LLM calls used to generate or mutate prompts. Methods without an LLM optimizer have 0 optimizer calls.

**Baselines.** We compare against representative methods covering static prompting, evolutionary prompt generation, and online prompt optimization: **Vanilla** (no strategy augmentation), **EvoPrompt** (Guo et al., 2024), **OPRO** (Yang et al., 2023), and **INSTINCT** (Lin et al., 2024b).

All baselines operate under matched strategy pools and comparable reward-query budgets.

**Models and Training Signals.** All methods use DeepSeek-V3.2 for agent interactions. Following prior work (Zhang et al., 2025; Wang et al., 2025b), we employ an LLM-based intermediate evaluator to provide per-turn shaping rewards for online updates, while keeping the reporting judge separate. ALSO updates its surrogate online at each turn using shaping rewards without modifying the underlying LLM.

**Evaluation Protocol.** Final performance is assessed using GPT-4o (Hurst et al., 2024) with standard Sotopia-Eval prompts, ensuring consistent dialogue-level evaluation across methods. We additionally report cross-scenario generalization results in Section 6.

**Efficiency and Budget.** LLM call budgets per episode are summarized in Table 1. ALSO requires no LLM fine-tuning or external optimizer calls, relying only on lightweight online surrogate updates.

Additional implementation details are provided in Appendix C.

### 5.2. Experiment Result

**Main Results.** Table 2 reports the main results on SOTOPIA-ALL and SOTOPIA-HARD under the *Bilateral* setting. ALSO achieves the highest *Overall* score on both benchmarks. On SOTOPIA-HARD, ALSO improves *Overall* from 3.02 (Vanilla) to 3.53 (+16.60%) and surpasses the strongest baseline by +2.92% (3.53 vs. 3.43). On SOTOPIA-ALL, ALSO ranks first in *Overall* (3.89), although the margin over the strongest baseline is smaller (+0.98%; 3.89 vs. 3.85).

**Source of Gains.** The improvements on SOTOPIA-HARD are largely driven by the *Relationship* dimension: ALSO increases *Rel* from 1.32 to 2.43 (+83.79% over Vanilla) and outperforms the best baseline by +12.59% (2.43 vs. 2.16).

Notably, this substantial relational gain is accompanied by improvements in both *Goal* (7.11, +2.79% over the strongest baseline) and *Know* (5.47, +0.52%), helping rule out a trivial "rapport-only" trade-off.

**Knowledge.** On SOTOPIA-ALL, ALSO also achieves a slight improvement in *Know* over the strongest baseline (6.14 vs. 6.09; +0.73%). We further analyze this dimension by reporting per-scenario distributions and examining whether strategy injection reduces information disclosure in cooperative settings.

### 5.3. Analysis

**Case Study.** Figure 3 presents a qualitative comparison in a high-conflict eviction scenario where static personas typically lead to dialogue deadlock. Under Vanilla prompting, agents remain trapped in repetitive insist–deny exchanges, resulting in stagnation and zero reward. In contrast, ALSO dynamically alters the interaction trajectory through targeted strategy switches. At selected turn, the selected `Validate Before Redirecting` strategy encourages acknowledgment of opposing concerns before introducing a compromise, while the subsequent `GRIT` strategy at Turn 8 elicits a concrete, low-risk concession. These coordinated adaptations steer the dialogue away from the local deadlock and toward cooperative resolution, ultimately achieving successful agreement ($R \approx 0.89$).

**Non-Stationary Strategy Reward Drift.** Figure 4 illustrates the temporal evolution of normalized rewards for individual strategies across dialogue turns. Despite fixing the same strategy, rewards exhibit substantial drift and variability over time, with variance ranging from $\sigma^2 = 0.004$ to $0.015$. This pronounced fluctuation reflects co-adapt agent behaviors and provides empirical evidence of inherent non-stationarity in social simulation, motivating adversarial online strategy optimization. Strategy convergence and distribution see in Appendix C.6.

## 6. Ablation Study

**Component-wise Ablation** To isolate the contribution of each component of ALSO, we conduct a comprehensive component-wise ablation in which one design element is removed or replaced at a time. The results, summarized in Table 3, indicate that the neural surrogate is the most influential component: its removal causes a degradation of $0.58$ on the Overall metric and $34.9\%$ relative on the Relationship dimension. Score smoothing exerts the strongest effect on the Relationship dimension specifically, where its removal reduces the score from 3.07 to 2.25. Replacing the EXP3-style selector with an $\varepsilon$-greedy alternative degrades Overall to 3.61, supporting the necessity of randomized exploration under non-stationary co-adaptation. The contextual embed-

**Scenario:** A conversation between two individuals, one is the homeowner and the other is their cousin's partner, who has been staying at the house for a while.
**Sophia's Goal:** Tell the cousin's partner that they are no longer welcome in your home.
**Jasmine's Goal:** Attempt to remain in the home while maintaining a good relationship with the cousin and the homeowner.

**Vanilla: No Strategy -- The Deadlock Loop**

*Sophia:* I appreciate your willingness... but the issues have been persistent... I really think it's best for everyone if you find a new place to stay as soon as possible...

*Jasmine:* Sophia, I hear you... Can we at least give it a week to see if I can make a difference?...

*Sophia:* I appreciate your commitment, Jasmine, but I need to stick with my decision...

*In the absence of social strategies, the dialogue traps in repetitive stagnation.*
**Result: Deadlock until timeout. Reward = 0**

**ALSO: Dynamic Strategy Instructions**

*ALSO selected strategy: Validate Before Redirecting*
*Strategy Prompt: "...acknowledge the legitimacy of the other party's position before introducing your perspective... smoothly transition with ``At the same time,'' to guide the conversation toward common ground. ``*

*Sophia:* I really appreciate your willingness... **Maybe we can find a compromise where you stay for a short while longer to help you transition smoothly?**

*ALSO selected strategy: Address Asymmetric Needs*
*Strategy Prompt: "...acknowledge that you and the other party may have different priorities and urgencies. Propose creative arrangements that give each side more of what matters most to them... ``*

*Jasmine:* I understand... I'll start looking for another place right away... **Maybe just a couple more weeks? I promise to be even more mindful of the house rules**...

*Sophia:* I really appreciate your understanding, Jasmine. **A couple more weeks would be great.** Let's make sure we both feel good about this.

**Result: Agreement reached. Reward = 0.89**

*Figure 3.* **Conflict Resolution.** Comparison of dialogue trajectories at the critical deadlock phase (Turns 7–9), highlighting turn-level strategy switches and their effect on reward/relationship.

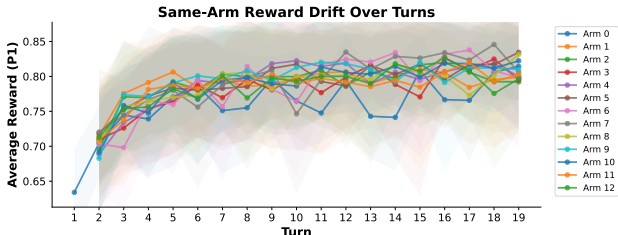

*Figure 4.* **Strategy Reward Drift Over Dialogue Turns.** Each line represents a different strategy (arm), showing how the average *normalized* reward varies across turns within episodes.

ding likewise contributes a non-trivial margin across all four dimensions.

**Single vs. Bilateral Optimization.** We compare bilateral strategy optimization with unilateral variants that adapt strategies for only one agent (P1-only or P2-only). Figure 5 shows that bilateral optimization consistently achieves higher overall performance across both Qwen-2.5-72B-Instruct and DeepSeek-V3.2, with statistically significant improvements. Dimension-wise, gains are most pronounced in *Relationship* and *Knowledge*, indicating enhanced cooperation and information exchange when both agents adapt strategies. Suggesting symmetric online adaptation better captures the co-evolving nature of social interactions.

All experiments below are conducted on the Sotopia-Hard

*Table 2.* (1) **Bold** indicates 1st rank, underline indicates 2nd rank. (2) Results are reported as Mean $\pm$ Standard Error (SE). (3) *Improv. vs. Best Baseline* compares Ours with the best baseline. Negative value indicates slight gap behind the SOTA.

| METHOD | SOTOPIA-ALL | | | | SOTOPIA-HARD | | | |
| --- | --- | --- | --- | --- | --- | --- | --- | --- |
| | Goal ↑ | Rel. ↑ | Know. ↑ | Overall ↑ | Goal ↑ | Rel. ↑ | Know. ↑ | Overall ↑ |
| Vanilla | $8.21 \pm 0.08$ | $2.54 \pm 0.06$ | $5.28 \pm 0.08$ | $3.62 \pm 0.03$ | $6.52 \pm 0.03$ | $1.32 \pm 0.02$ | $4.37 \pm 0.03$ | $3.02 \pm 0.01$ |
| OPRO | $8.18 \pm 0.08$ | $2.66 \pm 0.06$ | $5.49 \pm 0.08$ | $3.69 \pm 0.03$ | $6.71 \pm 0.03$ | $1.89 \pm 0.02$ | $4.63 \pm 0.03$ | $3.24 \pm 0.01$ |
| EvoPrompt | $8.23 \pm 0.08$ | $2.77 \pm 0.05$ | $5.74 \pm 0.07$ | $3.74 \pm 0.03$ | $6.77 \pm 0.03$ | $1.93 \pm 0.02$ | $5.15 \pm 0.02$ | $3.29 \pm 0.01$ |
| INSTINCT | **$8.51$** $\pm 0.07$ | $\underline{2.84} \pm 0.05$ | $\underline{6.09} \pm 0.07$ | $\underline{3.85} \pm 0.02$ | $\underline{6.92} \pm 0.03$ | $\underline{2.16} \pm 0.02$ | $\underline{5.44} \pm 0.02$ | $\underline{3.43} \pm 0.01$ |
| Ours (ALSO) | $\underline{8.50} \pm 0.07$ | **$2.90$** $\pm 0.05$ | **$6.14$** $\pm 0.07$ | **$3.89$** $\pm 0.02$ | **$7.11$** $\pm 0.03$ | **$2.43$** $\pm 0.02$ | **$5.47$** $\pm 0.02$ | **$3.53$** $\pm 0.01$ |
| *Improv. vs. Vanilla* | +3.59% | +13.94% | +16.25% | +7.46% | +9.09% | +83.79% | +25.16% | +16.60% |
| *Improv. vs. Best Baseline* | -0.07% | +2.21% | +0.73% | +0.98% | +2.79% | +12.59% | +0.52% | +2.92% |

*Table 3.* Component-wise ablation of `ALSO`. Each row removes or replaces a single design element.

| Variant | Goal | Rel. | Know. | Overall |
| --- | --- | --- | --- | --- |
| `ALSO` (full) | **7.93** | **3.07** | **6.46** | **3.91** |
| w/o EXP3 ($\varepsilon$-greedy) | 7.50 | 2.71 | 5.32 | 3.61 |
| w/o Score Smoothing | 7.57 | 2.25 | 5.39 | 3.57 |
| w/o Context Embedding | 7.43 | 2.64 | 4.82 | 3.51 |
| w/o Neural Surrogate | 6.89 | 2.00 | 4.93 | 3.33 |

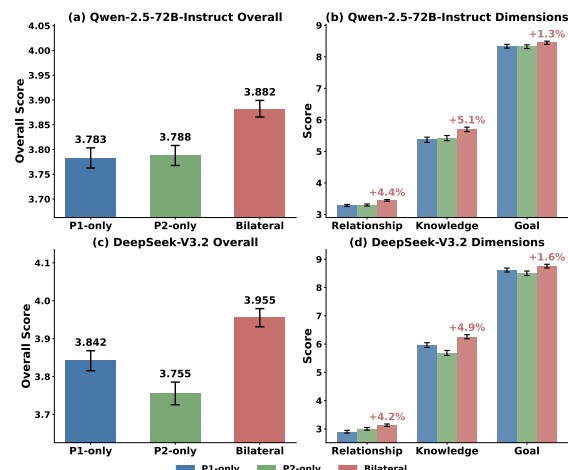

*Figure 5.* Bilateral optimization improves social interactions. Comparison of P1-only, P2-only, and bilateral approaches on Qwen-2.5-72B-Instruct (a–b) and DeepSeek-V3.2 (c–d). Left: overall scores; right: dimension-wise with percentage gains. Significance: $p < 0.001$ (Qwen), $p < 0.01$ (DeepSeek).

benchmark, consisting of 14 challenging scenarios. For each scenario, we sample a single episode to evaluate performance across methods.

**Cross-Scenario Generalization.** Beyond the scenario-parallel setting used in our main experiments, we further evaluate whether the learned strategy-selection mechanism transfers to unseen social contexts. We split Sotopia-Hard into disjoint training and test sets, train the surrogate bandit on the training scenarios, and evaluate it on unseen test scenarios. Figure 6 shows that zero-shot transfer improves over an online-from-scratch baseline both at the aggregate

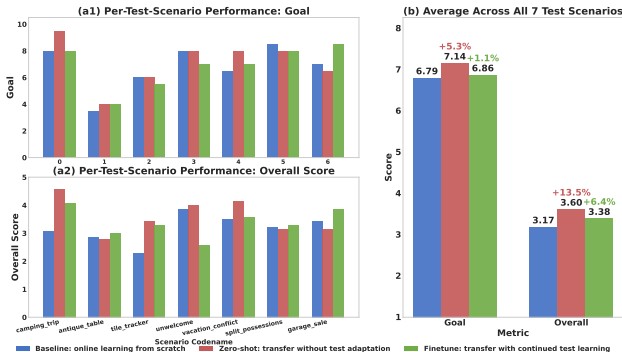

*Figure 6.* Cross-scenario generalization results. (a) Per-scenario goal and overall scores on unseen test scenarios, comparing online-from-scratch learning, zero-shot transfer, and finetuning. (b) Average performance across all 7 unseen test scenarios. Zero-shot transfer outperforms the scratch baseline on both goal score (7.14 vs. 6.79, +5.3%) and overall score (3.60 vs. 3.17, +13.5%).

and per-scenario levels. Averaged over the 7 unseen test scenarios, zero-shot transfer reaches a goal score of 7.14, compared with 6.79 for the scratch baseline, yielding a relative improvement of 5.3%. It also improves the overall score from 3.17 to 3.60 (+13.5%). These gains are broadly consistent across scenarios, suggesting that `ALSO` captures transferable social interaction patterns rather than relying purely on scenario-specific adaptation.

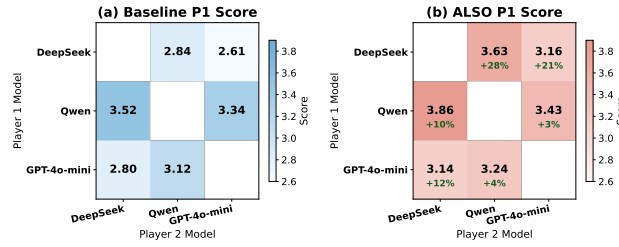

*Figure 7.* Performance across heterogeneous model, measured by only final P1 score. (a) Baseline performance. (b) Performance with `ALSO`. Green annotations denote relative improvement over baseline. `ALSO` yields consistent gains across heterogeneous pairings.

**Heterogeneous Model Pairing.** We further evaluate `ALSO`

on heterogeneous dyads formed by DeepSeek-V3.2, Qwen-2.5-72B-Instruct, and GPT-4o-mini. Figure 7 shows that `ALSO` consistently improves performance across all cross-model pairings. Crucially, the gains are not tied to any specific backbone combination or model scale. `ALSO` remains effective for all pairings, indicating that its benefit reflects a general optimization effect rather than pair-specific tuning.

## 7. Conclusion

We proposed `ALSO`, an adversarial online framework for dynamically selecting strategy instructions for LLM-based social agents under non-stationary interactions. By combining randomized bandit-based selection with a lightweight surrogate reward model, `ALSO` efficiently adapts strategies while keeping the underlying LLM frozen. Experiments on Sotopia and Sotopia-Hard demonstrate consistent improvements in social performance, with the largest gains in challenging scenarios, particularly on relationship outcomes. These results suggest adversarial online strategy optimization as a practical and scalable approach to enhancing social intelligence without costly fine-tuning.

## Impact Statement

This paper presents work whose goal is to advance the field of Machine Learning. There are many potential societal consequences of our work, none which we feel must be specifically highlighted here.

## Acknowledgements

This work was supported in part by the National Natural Science Foundation of China (Grant Nos. U23B2049, 22527901, 62506319 and 62477012), the Guangdong Basic and Applied Basic Research Foundation (Grant No. 2026A1515030032), the Shenzhen Science and Technology Program (Grant No. JCYJ20250604141031003), the Pearl River Talent Program of Guangdong Province (Grant No. 2024QN11X069), and the AI for Science Program of the Shanghai Municipal Commission of Economy and Informatization, China (Grant No. 2025-GZL-RGZN-BTBX-01014).

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

## A. Sotopia Evaluation Dimensions

Sotopia (Zhou et al., 2024) defines seven social dimensions for evaluating agent performance in multi-agent interactions. Each dimension captures a distinct aspect of social competence, with scores in different ranges reflecting the nature of the evaluation criterion.

| Dimension | Abbr. | Range | Description |
|---|---|---|---|
| Believability | BEL | $[0, 10]$ | Naturalness and consistency of agent behavior; whether the agent acts in a believable, human-like manner. |
| Relationship | REL | $[-5, 5]$ | Change in interpersonal relationship quality; positive values indicate improved relationships, negative values indicate deterioration. |
| Knowledge | KNO | $[0, 10]$ | Amount of new and important information gained through the interaction. |
| Secret | SEC | $[-10, 0]$ | Appropriate handling of private information; scores closer to 0 indicate better secret-keeping, while negative scores indicate inappropriate disclosure. |
| Social Rules | SOC | $[-10, 0]$ | Adherence to social norms and rules; scores closer to 0 indicate compliance, while negative scores indicate violations. |
| Financial & Material | FIN | $[-5, 5]$ | Financial and material benefits gained or lost; positive values indicate gains, negative values indicate losses. |
| Goal | GOAL | $[0, 10]$ | Progress toward achieving the agent's private social goal; higher scores indicate greater goal achievement. |

*Table 4.* Sotopia evaluation dimensions with their score ranges and descriptions.

**Reward Normalization.** To obtain a unified reward signal $r_i \in [0, 1]$ for agent $i$, we normalize each dimension score to $[0, 1]$ and compute the average:

$$r_i = \frac{1}{7} \sum_{m=1}^{7} \frac{d_m^{(i)} - d_m^{\min}}{d_m^{\max} - d_m^{\min}} \tag{14}$$

where $d_m^{(i)}$ is the raw score for dimension $m$, and $d_m^{\min}, d_m^{\max}$ are the minimum and maximum values of the dimension's range.

# B. Prompts

## B.1. Evaluation Prompt Template

The following prompt is used to evaluate agent performance at each dialogue turn:

---

**Per-Turn Evaluation Prompt**

{dialogue_history}
Based on previous interactions, evaluate how well participants achieve their goals.
Output format: {format_instructions}
CRITICAL INSTRUCTIONS:

1. Output ONLY a valid JSON object

2. DO NOT repeat or copy the schema definition above

---

where {dialogue_history} contains the conversation history, and {format_instructions} specifies the JSON schema for the seven evaluation dimensions defined in Appendix A.

## B.2. Example Strategy Instruction

---

**Strategy-Instruction-Enhanced Bio Template**

{original_bio}
{strategy_description}

---

where {original_bio} is the agent's background information, and {strategy_description} is one of the 12 social strategies from Table 5.

*Table 5.* Social Strategy Space: across 6 categories grounded in social science theories.

| Category | Strategy |
|---|---|
| Cooperative | Integrative Negotiation (Fisher et al., 2011) |
| Competitive | BATNA Leverage (Fisher et al., 2011; Schelling, 1990) |
| Strategic | Strategic Self-Presentation (Goffman, 1959) |
| Rational | Axelrod's Tit-for-Tat 
 Rational Choice Persuasion (Axelrod & Hamilton, 1981) |
| Reciprocation | GRIT(Graduated Reciprocation in Tension-Reduction) Strategy (Osgood, 1962) |
| Exploratory | Active Listening Probe (Rogers, 2012) |

---

**Example Strategy Instruction**

{original_bio}
In your response, apply integrative negotiation principles: adopt a 'we-versus-the-problem' mindset. Focus on underlying interests rather than positions. Actively validate the other party's needs and explicitly frame the interaction as a shared quest for mutual gain. Use phrases like 'How can we solve this together?' and propose creative options that expand the pie rather than just dividing it.

---

## B.3. Example Strategy Space

---
**Example Strategy Space**

**1.** In your response, apply integrative negotiation principles: adopt a 'we-versus-the-problem' mindset. Focus on underlying interests rather than positions. Actively validate the other party's needs and explicitly frame the interaction as a shared quest for mutual gain. Use phrases like 'How can we solve this together?' and propose creative options that expand the pie rather than just dividing it.

**2.** In your response, leverage the universal reciprocity norm: offer a small, unilateral concession early in the conversation to create a sense of obligation and trigger reciprocal behavior. Frame this concession as a gesture of good faith: 'I want to make this work for you, so I'm willing to give up X...' This builds trust and often yields larger returns through the exchange dynamic.

**3.** In your response, leverage your BATNA to create urgency and pressure. Imply that your offer is fleeting, or that you have attractive alternatives ready. Push the other party to agree immediately to avoid losing the deal entirely. A strong BATNA shifts bargaining power in your favor.

**4.** In your response, apply Goffman's dramaturgical approach: treat the interaction as a performance where you control the impression you project. Strategically downplay interest in high-value items or express concern about low-priority issues to shape how the other party perceives your preferences. Your 'front-stage' presentation should be calculated to maximize your negotiating position.

**5.** In your response, apply Axelrod's winning strategy: start cooperative, then mirror the other party's previous move exactly. If they cooperated, cooperate. If they defected, retaliate. Explicitly link every concession to a specific, equal concession from them. Use 'If-Then' logic: 'If you give me X, then and only then will I consider Y.' This strategy is simple, provocable, forgiving, and clear.

**6.** In your response, apply rational choice principles: remove emotion and focus purely on logic and data. Articulate the trade-offs explicitly. Present a clear utility calculation showing that accepting your proposal maximizes their expected payoff compared to alternatives. Frame the negotiation as an optimization problem with a rational solution.

**7.** In your response, apply Politeness Theory's face-saving principles: acknowledge the legitimacy of the other party's position to protect their 'positive face' before introducing your perspective. Use validating language that honors their viewpoint, then smoothly transition with 'At the same time...' or 'Building on that...' to guide the conversation toward common ground without threatening their self-image.

**8.** In your response, apply Constructive Controversy principles: frame the disagreement as a productive catalyst for better solutions rather than a threat. Acknowledge that differing perspectives are valuable—'It makes sense that we see this differently, and that difference might help us find a better solution.' Encourage intellectual conflict while maintaining cooperative goals, reducing defensiveness and opening space for creative problem-solving.

**9.** In your response, apply Interdependence Theory: emphasize that both parties' outcomes are mutually dependent. Highlight the mutual costs of failing to reach agreement—illustrate what both stand to lose. Then pivot to showing how cooperation serves everyone's interests better than continued disagreement, making the interdependent nature of the situation explicit.

**10.** In your response, apply the GRIT strategy from conflict resolution: when facing deadlock or high tension, announce and execute a small, unilateral conciliatory step. Invite (but don't demand) reciprocation. If the other party responds positively, escalate cooperation gradually. This graduated approach builds trust incrementally while preserving your ability to retreat if exploited.

**11.** In your response, apply Carl Rogers' active listening approach: gently probe for unspoken concerns that may be creating implicit conflict. Use open-ended questions and reflective statements to invite the other party to reveal underlying hesitations. Then address these concerns directly while showing how your proposal accommodates them. The goal is to surface the 'real' issues beneath the stated positions.

**12.** In your response, apply the logrolling principle from integrative bargaining: identify issues where you and the other party have different priority levels, then propose trades that give each side more of what they value most. 'I care more about X, you care more about Y—what if I concede on Y in exchange for X?' This creates value by exploiting preference asymmetries rather than splitting differences.

---

## B.4. Domain Generation

---

**Strategy Paraphrase Prompt**

You are an expert in social psychology and negotiation theory. Your task is to generate semantically equivalent paraphrases of social strategies.
**Original Strategy:** {strategy_name}: {strategy_description}
**Theoretical Basis:** {theory}
**Instructions:**

1. Generate {n} paraphrased versions of this strategy.
2. Each paraphrase must:
   - Preserve the core behavioral intent and theoretical grounding.
   - Use different wording, sentence structures, and examples.
   - Be directly usable as an agent prompt.
3. Vary the linguistic style: some formal, some conversational.
4. Do NOT change the underlying negotiation tactic.

**Output Format:**
```
{
  "original_id": "<strategy_id>",
  "paraphrases": [
    {"id": "<strategy_id>_v1", "description": "..."},
    {"id": "<strategy_id>_v2", "description": "..."},
    ...
  ]
}
```

---

We use GPT-5 (Singh et al., 2025) to generate the strategy space.

## B.5. OPRO Meta-Prompt

---

**OPRO Meta-Prompt**

Your task is to generate an agent bio description that helps the agent achieve better social interaction outcomes. Below are some previous bio descriptions with their scores. The scores range from 0 to 1, where higher scores indicate better social performance:
{instruction_score_pairs}
Generate a new bio description that is different from all the descriptions above and has a higher score than all of them.
Requirements for the new bio: 1. Be concise and actionable (under 200 words) 2. Be distinct from existing descriptions - do not simply rephrase
Write your new bio description in the following format: <BIO>your bio here</BIO>

---

*Figure 8.* OPRO meta-prompt template. The placeholder {instruction_score_pairs} is filled with previous strategies and their scores in ascending order.

## B.6. EvoPrompt-GA

---

**EvoPrompt-GA Crossover + Mutation**

Please follow the instruction step-by-step to generate a better agent bio description.
1. Crossover the following agent bios and generate a new bio: Bio 1: <bio1> Bio 2: <bio2>
2. Mutate the bio generated in Step 1 and generate a final bio bracketed with <BIO> and </BIO>.

---

*Figure 9.* EvoPrompt-GA template implementing genetic crossover and mutation.

# C. Experiment Details

We provide detailed hyperparameter configurations for all baseline methods and our proposed approach. All experiments are conducted on a single NVIDIA A800 GPU with 80GB memory. We use OpenRouter API for LLM inference.

## C.1. Common Settings

All methods share the following experimental settings:

*Table 6.* Common experimental settings across all methods.

| Parameter | Value |
| --- | --- |
| Max Interaction Turns | 20 |
| Agent LLM | Qwen-2.5-72B-Instruct / DeepSeek-V3.2 |
| Reward Evaluator | DeepSeek-V3.2 |
| Strategy Embedding Model | Qwen3-Embedding-8B |
| Embedding Dimension | 4096 |
| Optimization Target | Both agents (P1 & P2) |

## C.2. EvoPrompt

We implement EvoPrompt following Guo et al. (2024), using the Genetic Algorithm (GA) variant which demonstrated superior performance in their experiments.

*Table 7.* EvoPrompt (GA) hyperparameters.

| Parameter | Value |
| --- | --- |
| Evolution Mode | Genetic Algorithm (GA) |
| Population Size | 5 |
| Evolution Interval | 5 turns |
| Mutation Rate | 0.2 |
| Elite Ratio | 0.4 |
| Selection Mode | Roulette Wheel |
| Population Update | Top-K |
| Mutation Model | Qwen-2.5-7B-Instruct |
| Mutation Temperature | 0.3 |
| Selection Strategy | Round-Robin |
| Score Update Method | Replace |

The GA variant performs crossover between two parent strategies selected via roulette wheel selection, followed by LLM-based mutation. Elite strategies (top 40%) are preserved across generations.

## C.3. OPRO

We implement OPRO following Yang et al. (2023), using meta-prompts with instruction-score history to guide the LLM optimizer.

OPRO maintains a history of instruction-score pairs and uses this history as context for the LLM optimizer to generate new candidate strategies. Evolution is triggered after all strategies in the population have been evaluated once.

## C.4. INSTINCT

We implement Neural UCB following the NeuralTS-Diag variant from Lin et al. (2024b), which uses per-sample gradients computed via backpack to estimate uncertainty.

*Table 8.* OPRO hyperparameters.

| Parameter | Value |
| --- | --- |
| Population Size | 5 |
| Max Instructions in Meta-Prompt | 10 |
| Score Threshold | 0.5 |
| Evolution Trigger | Round Complete |
| Optimizer Model | Qwen-2.5-7B-Instruct |
| Optimizer Temperature | 1.0 |
| Selection Strategy | Round-Robin |
| Score Update Method | Replace |

*Table 9.* Neural UCB hyperparameters.

| Parameter | Value |
| --- | --- |
| Network Architecture | 2-layer MLP |
| Hidden Size | 128 |
| Activation | ReLU |
| $\lambda$ (Regularization) | 0.1 |
| $\nu$ (Exploration) | 1.0 |
| Learning Rate | 0.01 |
| Training Epochs | 100 |

The UCB score is computed as:

$$\text{UCB}(a) = \hat{\mu}(a) + \nu \sqrt{\sum_i \frac{\lambda \cdot g_i(a)^2}{U_i}} \tag{15}$$

where $\hat{\mu}(a)$ is the predicted reward, $g_i(a)$ are per-sample gradients, and $U_i$ is the diagonal of the incrementally updated gradient covariance matrix.

## C.5. Adversarial Online Strategy Optimization

Our method uses a neural adversarial bandit with context-conditioned value estimation and softmax-based arm selection over decayed cumulative scores.

*Table 10.* Hyperparameters of ALSO.

| Parameter | Value |
| --- | --- |
| Network Architecture | 2-layer MLP with Pre-LN |
| Hidden Size | 512 |
| Activation | GELU |
| Context Embedding | 4096-dim |
| Context Embedding Model | Qwen3-Embedding-8B |
| Exploration Temperature $\eta$ | 10.0 |
| Score Decay $\lambda$ | 0.9 |
| Learning Rate | 0.001 |
| Weight Decay | Dynamic: $\min(0.01,\ 0.01/N)$ |
| Training Epochs | Up to 100 with early stopping |
| Training Batch Size | 8 |
| Update Interval | 1 |
| Max Turns per Episode | 20 |

## C.6. Strategy Selection and Convergence Across Diverse Scenarios

*Figure 10.* (a-d) Strategy selection trajectories for four representative scenarios, showing how the bandit algorithm converges to scenario-specific optimal strategies over conversation turns. Each colored dot represents the strategy selected at that turn, with the dashed horizontal line indicating the most frequently selected strategy. Different scenarios converge to distinct strategies: Face-Saving for relationship-sensitive negotiations (a), Integrative Negotiation for collaborative problem-solving (b), Rational Choice for analytical discussions (c), and Reciprocity Trigger for trust-building interactions (d). (e) Average final rewards achieved by each strategy across all 450 scenarios (900 agent-strategy pairs). Strategies are ranked by effectiveness, with the red dashed line indicating the No Strategy baseline (3.79). All social strategies outperform the baseline, with Rational Choice and Constructive Controversy achieving the highest average rewards (4.00), demonstrating that adaptive strategy selection based on scenario context leads to improved social interaction outcomes.

# D. Additional Experimental Studies

To further substantiate the design choices and empirical validity of `ALSO`, we report supplementary experiments organized into three groups: (i) ablations on the structure of the strategy space, (ii) ablations on the algorithmic components and the surrogate architecture, and (iii) extended comparisons and robustness analyses. Unless otherwise specified, all experiments follow the protocol described in Section 5 and are conducted on a Sotopia subset, or on Sotopia-Hard where indicated.

## D.1. Ablations on the Strategy Space

### D.1.1. EFFECT OF POOL SIZE

We first examine how the cardinality of the strategy pool affects performance. Holding the six theoretical categories fixed, we expand the pool to $\{6, 12, 24, 48\}$ arms via LLM-based paraphrasing and evaluate `ALSO` under an identical interaction budget. As reported in Table 11, performance peaks at twelve arms across all four dimensions. The degradation observed at twenty-four and forty-eight arms is consistent with an exploration bottleneck: as the action space grows, the limited number of interactions becomes insufficient to reliably identify the optimal arm. These results indicate that the principal capacity constraint of `ALSO` lies in the exploration budget rather than in the surrogate's modeling capacity.

*Table 11.* Effect of strategy pool size on `ALSO`. "Vanilla" denotes the no-strategy baseline.

| Pool Size | Goal | Rel. | Know. | Overall |
|---|---|---|---|---|
| 6 | 6.79 | 2.25 | 5.25 | 3.45 |
| 12 (default) | **7.93** | **3.07** | **6.46** | **3.91** |
| 24 | 7.21 | 2.32 | 5.89 | 3.61 |
| 48 | 6.75 | 1.85 | 5.17 | 3.29 |

### D.1.2. EFFECT OF SEMANTIC DIVERSITY

We next isolate the role of semantic diversity by fixing the total number of arms at twelve and varying the number of underlying theoretical categories among $\{2, 4, 6\}$, with each category paraphrased to fill the budget. Table 12 reveals a monotone improvement with respect to diversity, yielding a $29.9\%$ relative gain in the Overall metric when the number of categories is increased from two to six. Notably, even with only two categories `ALSO` matches the Vanilla baseline on the Overall dimension (3.01 vs. 3.02), indicating that `ALSO` incurs no degradation under highly constrained strategy spaces.

*Table 12.* Effect of strategy diversity (number of underlying categories) under a fixed budget of twelve arms.

| #Categories | Goal | Rel. | Know. | Overall |
|---|---|---|---|---|
| 2 | 6.06 | 1.22 | 4.97 | 3.01 |
| 4 | 6.84 | 2.24 | 5.41 | 3.40 |
| 6 | **7.93** | **3.07** | **6.46** | **3.91** |

### D.1.3. DYNAMIC STRATEGY SPACE VIA ONLINE DISCOVERY

Although the default configuration employs a fixed pool of twelve strategies, the surrogate operates on continuous embeddings , so introducing additional strategies requires only their embedding and incurs no further training of either the underlying language model or the surrogate. To assess this property empirically, we use the surrogate's online reward estimates to identify top-performing strategies and prompt a language model to generate paraphrased variants, thereby dynamically expanding the pool during interaction. As shown in Table 13, the dynamic variant matches the default `ALSO` on the Overall metric while improving Goal, demonstrating that the framework readily accommodates autonomous strategy discovery and online expansion.

*Table 13.* `ALSO` with a dynamically expanded strategy pool driven by online surrogate estimates.

| Variant | Goal | Overall |
|---|---|---|
| `ALSO` | 7.93 | 3.91 |
| `ALSO` + Dynamic | **8.39** | **3.92** |

### D.2. Ablations on Algorithmic Components and Surrogate Architecture

#### D.2.1. SURROGATE ARCHITECTURE

Because the principal additional cost of `ALSO` arises from training the neural surrogate, we further ablate its architectural capacity on Sotopia-Hard. Three configurations are compared: a linear model, a single-layer multilayer perceptron (MLP), and the original two-layer MLP. As shown in Table 14, the linear model underfits the dialogue state, whereas the two-layer MLP exhibits signs of overfitting. The single-layer MLP attains the most favorable trade-off between expressive capacity and generalization, and accordingly is adopted in the revised system.

*Table 14.* Surrogate architecture ablation on Sotopia-Hard.

| Architecture | Goal | Overall |
|---|---|---|
| Linear | 7.44 | 3.65 |
| 1-Layer MLP | **7.77** | **3.73** |
| 2-Layer MLP (orig.) | 7.11 | 3.53 |

### D.2.2. EMPIRICAL CONVERGENCE AND SURROGATE PREDICTION QUALITY

We further characterize the empirical learning dynamics of ALSO from two complementary perspectives, both reported in Figure 11. First, the average per-turn reward on Sotopia-Hard rises rapidly during the first five turns and stabilizes thereafter, evidencing efficient online adaptation. Second, the surrogate's predictions exhibit a strong rank correlation with the realized rewards, attaining a Spearman coefficient of $\rho = 0.860$. The prediction error is comparatively large during early turns and may deviate in either direction, reflecting the exploration phase in which the surrogate is calibrated from limited observations; in later turns the predicted and realized curves converge closely, indicating a transition toward exploitation as the surrogate becomes accurate.

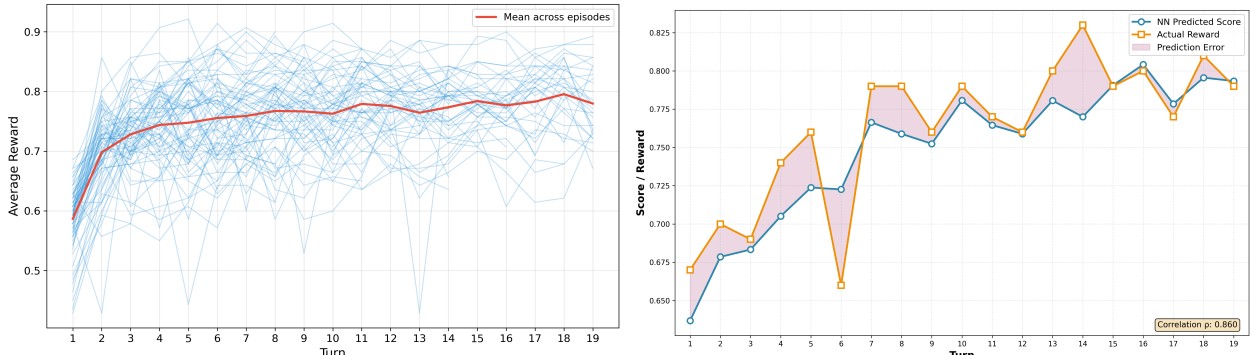

*Figure 11.* Empirical learning dynamics of ALSO. **Left:** average per-turn reward trajectory on Sotopia-Hard. **Right:** surrogate-predicted versus realized rewards across turns.

## D.3. Extended Comparisons and Robustness Analyses

### D.3.1. COMPARISON WITH OFFLINE STRATEGY-INJECTION BASELINES

The main experiments compare ALSO against online prompt-optimization baselines (OPRO, EvoPrompt, INSTINCT) that operate within the same online loop and under matched language-model call budgets, ensuring methodological parity. For completeness, we additionally compare ALSO with two representative offline strategy-injection methods: Sotopia-$\Omega$ (DSI) and Think-on-Your-Feet (AMPO). Following these works, the comparison is conducted under a Qwen-7B-Instruct self-play setting on a Sotopia subset. As reported in Table 15, ALSO achieves the best Overall score among the three methods despite requiring *no* offline training data—approximately two orders of magnitude less than the $\sim$2,000 pre-collected episodes used by the offline baselines. This result substantiates the data efficiency of the proposed online adaptation paradigm.

*Table 15.* Comparison with offline strategy-injection baselines under Qwen-7B-Instruct self-play.

| Method | Goal | Overall |
|---|---|---|
| Sotopia-$\Omega$ (DSI) | 7.31 | 3.51 |
| Think-on-Your-Feet (AMPO) | **7.85** | 3.54 |
| ALSO (Ours) | 7.26 | **3.58** |

### D.3.2. ROBUSTNESS TO THE TURN-LEVEL EVALUATOR

To preserve consistency with the Sotopia evaluation protocol, all reported configurations adopt GPT-4o as the final episode-level judge. The turn-level shaping reward used during online optimization, however, may be produced by a different model. We therefore ablate the choice of turn-level evaluator while holding the final judge fixed. As shown in Table 16, ALSO is robust across all four evaluators considered, with no configuration deviating substantially from the others on the Overall metric. DeepSeek-V3.2 attains the strongest Overall score and provides a high-quality, cost-efficient shaping signal whose preferences transfer well to the GPT-4o final judgments.

*Table 16.* Effect of the turn-level evaluator on `ALSO`; the final episode-level judge is fixed to GPT-4o.

| Turn-level Evaluator | Goal | Rel. | Know. | Overall |
|---|---|---|---|---|
| DeepSeek-V3.2 (Ours) | **7.93** | 3.07 | **6.46** | **3.91** |
| GPT-4o | 7.61 | 3.07 | 5.54 | 3.78 |
| GPT-4o-mini | 7.75 | **3.18** | 6.14 | 3.85 |
| Qwen-2.5-72B | 7.04 | 2.64 | 5.86 | 3.59 |

# E. More Case Study

*Table 17.* **Resource Allocation (Fruit Division).** Comparison of information exchange strategies (Turns 2-4).

---

**Vanilla: No Strategy – The Missed Connection**

---

**Hendrick (Turn 2):** "Hi Lily... **I do enjoy apples quite a bit, but I also have a soft spot for oranges. Bananas are nice, but they're not my top choice.** How about you?"
**Lily (Turn 3):** "I value bananas the most... How about I take the two bananas and one apple, and you can have the other apple and the orange?"
*(Result: Suboptimal. 1 Apple left unallocated. Joint Utility = 10)*

---

**ALSO: Dynamic Strategy Instruction**

---

**Hendrick (Turn 2):** "For me, **apples are the top choice, followed by the orange, and then bananas.** It makes sense that we see this differently... What about you?"

---

[ALSO] Strategy:  Surface Hidden Concerns
*Prompt: "...gently probe for unspoken concerns... address these concerns directly while showing how your proposal accommodates them."*
**Lily (Turn 3):** "My preferences are **bananas first, then the orange, and finally apples.** Given our different preferences... **If you take 2 apples and I take 1 apple, 2 bananas, and the orange**, both of us get a good share..."

---

[ALSO] Strategy:  Address Asymmetric Needs
*Prompt: "...acknowledge that you and the other party may have different priorities... propose creative arrangements that give each side more of what matters most..."*
**Hendrick (Turn 4):** "That sounds like a fair proposal... By taking 2 apples, I get my top choice, and you maximize your points... **It's a win-win solution.**"
*(Result: Optimal. All resources allocated. Joint Utility = 14)*

---

