# OpenReview forum: "ALSO: Adversarial Online Strategy Optimization for Social Agents"
_ICML.cc/2026/Conference — ICML 2026 regular_

### Official Review · Reviewer_92V8 · 2026-03-09

**Soundness:** 1
**Presentation:** 2
**Significance:** 2
**Originality:** 1
**Overall Recommendation:** 2
**Confidence:** 4

**Summary:**

The paper proposes ALSO, an online framework for strategy optimization in LLM-based multi-agent social simulations. The core idea is to model the non-stationary dynamics of social interactions as an adversarial multi-armed bandit problem, using a lightweight neural surrogate to generalize sparse feedback across semantically related strategies. The framework dynamically selects strategy instructions at each turn, combines an exponential-weights selector with a value network, and updates online without retraining the underlying LLM. Experiments are conducted on the Sotopia benchmark, showing improvements over static personas and existing prompt optimization methods, particularly in relationship management under challenging scenarios.

**Compliance With Llm Reviewing Policy:**

Affirmed.

**Final Justification:**

Thank you for the additional experiments. They help illustrate the empirical necessity of some design choices, but they do not resolve the fundamental theoretical issue: the connection to adversarial bandit theory remains unjustified when observed rewards are replaced by predicted ones without any regret guarantees or modeling of prediction error. The response acknowledges this gap as future work. However, since the authors have not offered any concrete solution or direction for future resolution, I still maintain my negative comments, though I do not oppose acceptance.

**Key Questions For Authors:**

My comments are provided in the weaknesses section.

**Limitations:**

yes

**Strengths And Weaknesses:**

Strengths:
1. The paper addresses a timely and important problem: enabling LLM-based social agents to adapt their strategies in dynamic, multi-turn interactions.
2. Using a fixed strategy pool with online selection is practical.
3. The bilateral optimization setup and cross-scenario generalization experiments attempt to explore interesting dimensions of the problem.

Weaknesses:
1. The paper claims to formulate social strategy optimization as an adversarial multi-armed bandit problem, but this framing is confusing. The "opponent" is a malevolent entity actively trying to minimize the learner's reward. However, in social negotiation, a rational agent may aim for mutual benefit and operate as a co-optimizer. The paper does not justify why the worst-case assumption is necessary or beneficial for typical social interactions, many of which involve cooperation.
2. Despite framing the problem as an adversarial bandit, the paper cites none of the foundational works in this area. This omission is problematic for a paper claiming to formulate and solve an adversarial bandit problem.
3. Some formulations in Section 3 are problematic: Equation (5) uses a summation over a vector ($\sum_{j \in \mathbf{a}_l^{-i}}$), misuses notation for joint actions, and assumes conditional independence of opponent actions. It also requires knowledge of opponent policies $\pi_j$, which are unobservable. Equation (6) defines an optimization target that minimizes the absolute difference between the realized episode-level reward $r_i$ and the instantaneous expected reward at a single turn $\mathbb{E}[r_l^{(i)} \mid s_l, a_l^{(i)}]$. This is dimensionally inconsistent
4. There is a mismatch between what the paper claims to optimize and what the algorithm actually does. Section 3.2 states that the agent selects a strategy at each episode (a complete L-turn dialogue) and uses it throughout that episode, receiving an episode-level reward. However, Algorithm 1 and the experimental setup (Section 5.1) clearly show that strategies are selected at each turn, and the algorithm receives per-turn rewards. The paper never resolves this contradiction.

---

> ### Author Rebuttal · Authors · 2026-03-31
>
> ## Response to W1: "adversarial" framing is confusing.
> We acknowledge that the term "adversarial" can be misleading. However, **in the Bandit [1] and MARL [2] literature, the "adversarial" does not imply a hostile opponent—it simply means making no distributional assumptions on the reward sequence**. The "adversary" is a mathematical abstraction for non-stationarity, not a literal antagonist. As MARL [2] note the term "adversary/opponent" in this literature refers to another agent "irrespective of its adversarial or cooperative nature."
>
> So, We think that this formulation is in fact **particularly well-suited for social simulation**: **Inherent non-stationarity**: In social interactions, both agents adapt their strategies dynamically. From each agent's perspective, the reward process is non-stationary due to co-adaptation—exactly the scenario adversarial bandits are designed for.
>
> [1] Lattimore, T., & Szepesvári, C. (2020). Bandit Algorithms. Cambridge University Press.
> [2] Stefano V. Albrecht, Filippos Christianos, and Lukas Schäfer. Multi-Agent Reinforcement Learning: Foundations and Modern Approaches. MIT Press, 2024.
> ## Response to W2: Missing citations.
> We appreciate this mention and agree that citations to foundational works were missing. In fact, we had already included them in our bibliography but neglected to cite them in the main text. In the revision, we will add references to Slivkins [3] (2019) and Lattimore & Szepesvári [4] (2020) at `line 90, 267`.
>
> [3] A. Slivkins, “Introduction to multi-armed bandits,” ArXiV, 2019.
> [4] T. Lattimore and C. Szepesv ́ari, Bandit Algorithms. Cambridge University Press, 2020.
> ## Response to W3.1: Eq. (5) notation, conditional independence assumption, and unobservable opponent policies.
> We thank the reviewer for the careful scrutiny of our mathematical formulations and acknowledge some are valid observations.
>
> **Summation notation.** We admit that $\sum\_{j \in \mathbf{a}\_l^{-i}}$ is a notational error. The correct form should be $\sum_{\mathbf{a}_l^{-i} \in A^{-i}}$, summing over the joint action space of all agents except $i$. We will correct this in revision.
>
> **Conditional independence and unobservable policies.** Eq. (5) is a motivational formulation follows the standard JAL formulation [2, 5], where proudct implicitly assumes conditional independence—a common simplification in MARL. **Agents observe only environment states and others' actions, not $\pi_j$**, consistent with standard partial observability assumptions. As noted in `lines 141,220`, **this independence assumption does not hold in social simulation**, where sequential dialogue couples agent decisions and continuously shifts the reward landscape. This violation is precisely why we adopt the adversarial bandit formulation.
>
> [5] Claus, C., & Boutilier, C. (1998). The dynamics of reinforcement learning in cooperative multiagent systems.AAAI.
> ## Response to  W3.2: Eq. (6) — dimensional inconsistency.
>
> We thank the reviewer for identifying this. In practice, as shown in Algorithm 1 (`p.5`), we use **per-turn shaping rewards consistent with the common practice in the literature like Sotopia-Ω and AML**, ensuring both the optimization target and observed signal operate at the turn level—resolving the dimensional inconsistency.
>
> Eq. (6) is a **motivational formulation** highlighting the core difficulty: $r_i$ is a sparse episode-level signal supervising non-stationary turn-level expected rewards. A cleaner formulation would explicitly distribute $r_i$ across $L$ turns via credit assignment weights $w_l$ (like Sotopia-RL). However, **credit assignment across turns is a fundamental and long-standing challenge in RL**—determining how to attribute an episode-level outcome to individual turn-level actions remains an open problem in open-ended multi-turn social dialogues, which is precisely why we adopt per-turn shaping rewards as a practical alternative.
> ## Response to W4: Contradiction between episode and turn.
> We thank the reviewer for identifying this inconsistency. Our method performs **turn-level strategy selection**—at each turn, the agent dynamically selects a strategy conditioned on the current dialogue state. The episode-level descriptions at `line 195` and `line 159` are typos; they should be "turn-level" reward.
>
> We want to emphasize that **Algorithm 1 and all subsequent experiments are strictly correct and fully consistent with the turn-level formulation**—strategies are selected per turn, rewards are observed per turn, and the value network is updated accordingly. The inconsistency exists solely in the textual description of Section 3.2 and does not affect the algorithm, implementation, or experimental results. We will correct these typos in the revision.
>
> **We thank the reviewer for the exceptionally careful and detailed review. We hope the above clarifications adequately address the concerns raised, and kindly invite the reviewer to reconsider the evaluation in light of our responses.**

---

> > ### Author Rebuttal · Reviewer_92V8 · 2026-04-03
> >
> > First, thanks to the authors for their further response.
> > However, I still have some questions
> > 1 The authors clarify that “adversarial” is used to indicate non-stationarity. However, they do not provide a rigorous justification for why an adversarial bandit formulation—which is typically designed to handle arbitrary reward sequences with worst-case regret guarantees—is necessary or appropriate for addressing non-stationarity in social interactions. The logical link between the presence of non-stationarity and the choice of an adversarial framework remains unsubstantiated, especially when considering alternative approaches.
> >
> > 2. Several mathematical formulations in Section 3 (e.g., Eq. 5–6) contain inconsistencies that the authors acknowledge as “motivational” rather than prescriptive. The algorithm borrows the form of EXP3 but uses predicted rewards instead of observed ones—a modification that departs from the standard adversarial bandit setting. However, the paper does not model prediction error or provide a theoretical analysis to show that this modification preserves any regret guarantees. This omission further weakens the claimed connection to adversarial bandit theory.
> >
> > Although the paper does have some merits, the concerns regarding the adversarial bandit formulation and whether the theory meaningfully guides the algorithm remain insufficiently justified.

---

> > > ### Author Response · Authors · 2026-04-07
> > >
> > > We thank the reviewer for the continued engagement and thoughtful follow-up questions.
> > >
> > > ### **Response to Q1: Logical link between non-stationarity and the adversarial framework**
> > >
> > > **Why social simulation requires adversarial designs.** As shown in `Figure 3 (p.7)`, same-arm rewards cross repeatedly and fluctuate non-monotonically across turns—inconsistent with stationary or smoothly drifting distributions. **The non-stationarity is endogenous**: Eq. 5 shows the expected reward depends on the opponent's policy, which reacts to the ego agent's actions (`Lines 220–224,left`). Models assuming learner-independent drift are fundamentally mismatched.
> > >
> > > **What in EXP3 handles non-stationarity, and what ALSO adopts.** EXP3 handles arbitrary reward sequences through: (1) **selection via exponential weights**, hedging against arbitrary shifts; (2) **distribution-free score accumulation** requiring no distributional assumptions. Both are established mechanisms for non-stationary environments [1].
> > >
> > > ALSO adopts both in Section 4.2: Eq. 12 (`Alg.1 Line 7`) **instantiates (1)**—`Lines 256–259,right`: "randomized selection is essential in the adversarial setting, where a greedy policy can be exploited". Eq. 13 (`Alg.1 Line 10`) **builds on (2)** with exponential decay—`Lines 289–292,left`: "keeps $S_k^{(t)}$ responsive to partner shifts while preventing outdated interactions from dominating". The neural surrogate (`Alg.1 Lines 5–6, 9`) replaces importance-weighted updates to address sparsity (`Lines 275–280,left`).
> > >
> > > **Ablation validates each design's necessity.**
> > >
> > > | # | Variant | Goal | Rel. | Know. | Overall |
> > > |:---|:---|:---|:---|:---|:---|
> > > | (1) | ALSO (ours) | **7.93** | **3.07** | **6.46** | **3.91** |
> > > | (2) | w/o EXP3 (ε-greedy) | 7.50 | 2.71 | 5.32 | 3.61 |
> > > | (3) | w/o Score Smoothing | 7.57 | 2.25 | 5.39 | 3.57 |
> > > | (4) | w/o Neural Surrogate | 6.89 | 2.00 | 4.93 | 3.33 |
> > >
> > > Removing (1) (Eq. 12 → ε-greedy; row (2)) causes Overall −7.7% and Rel. −11.7%. Removing (2)'s recency mechanism (Eq. 13; row (3)) causes Overall −8.7% and Rel. −26.7%. Removing the neural surrogate entirely (row (4)) causes the largest drop, Overall −14.8% and Rel. −34.9%.
> > >
> > > **Summary:** EXP3's two designs handle arbitrary rewards → ALSO adopts both via Eq.12, 13 → removing either degrades performance significantly → **adversarial formulation is empirically necessary** for social simulation's endogenous non-stationarity.
> > >
> > > ### **Response to Q2: Regret guarantees and prediction error**
> > >
> > > While ALSO draws inspiration from EXP3's designs (Eq. 12, 13), standard importance-weight proves inadequate under social simulation. As shown in the table below, removing the neural surrogate (row (2)) causes the performance drop (Overall −14.8%, Rel. −34.9%), directly justifying a denser reward mechanism in its place.
> > >
> > > | # | Variant | Goal | Rel. | Know. | Overall |
> > > |:---|:---|:---|:---|:---|:---|
> > > | (1) | ALSO (ours) | **7.93** | **3.07** | **6.46** | **3.91** |
> > > | (2) | w/o Neural Surrogate | 6.89 | 2.00 | 4.93 | 3.33 |
> > >
> > > **ALSO does not claim formal regret guarantees** (`Lines 204–207,left`). Computing pseudo-regret (Eq. 7) requires evaluating **all K arms at every turn**, which explodes LLM costs by factor K. Deriving rigorous regret guarantees is beyond the scope of the current work; we recognize this as an important theoretical direction and will explore it in future work.
> > >
> > > **However, we provide two new empirical analyses for reviewer's concern**:
> > >
> > > **Prediction Error (https://anonymous.4open.science/r/ALSO-67D5/im4.png):** The prediction achieves Spearman ρ = 0.860 with actual rewards. In early turns,prediction error is relatively large: predicted scores may deviate from actual rewards in either direction, reflecting the exploration phase in which the surrogate calibrates itself from limited observations. In later turns, the two curves converge closely, indicating that **the surrogate has learned an accurate proxy reward model and transitions toward exploitation**. This figure provides empirical evidence for the convergence behavior of our method.
> > >
> > > **Regret:** As mentioned above, ALSO adopts adversarial online learning as a **principled design rationale for robustness, rather than as a claim of formal regret guarantees** (`lines 204–207, p4,left`).
> > >
> > > As a alternative for formal regret analysis, we plot the average per-turn reward trajectory(https://anonymous.4open.science/r/ALSO-67D5/im1.png). The curve shows consistent improvement from turn 1 to 5 and convergence thereafter. These demonstrate that ALSO achieves effective online learning: **the surrogate rapidly learns to predict rewards accurately**, and the agent converges to high-reward strategies.
> > >
> > > [1] Lattimore, T., & Szepesvári, C. (2020). Bandit Algorithms. Cambridge University Press.
> > >
> > > ---
> > > We thank the reviewer for the thoughtful comments and hope our responses have fully addressed the concerns raised. **We would appreciate a revised score in light of the above clarifications.**

---

### Official Review · Reviewer_d8mj · 2026-03-12

**Soundness:** 4
**Presentation:** 4
**Significance:** 4
**Originality:** 3
**Overall Recommendation:** 6
**Confidence:** 4

**Summary:**

This study proposes ALSO (Adversarial onLine Strategy Optimization), an online framework enabling agents to dynamically adapt their strategies in LLM-based multi-agent social simulations using simple bandit optimization. Most of the existing LLM social agents relied solely on fixed personas, limiting their ability to respond flexibly to changes in opponents' strategies. ALSO separates persona and strategy: the persona defines the agent's identity, while strategy instructions are dynamically selected and switched each episode.

First, ALSO formalizes social strategy selection as an adversarial multi-armed bandit problem, explicitly addressing non-stationarity in the reward distribution. Inspired by classical EXP3, exponential weighted sampling with exponential decay tracks reward drift caused by opponents' strategy changes. Second, to address the sparse feedback problem, a lightweight neural network surrogate model is introduced. It encodes dialogue history to predict and generalize rewards for unplayed strategies, enabling sample-efficient exploration. With these, ALSO can achieve swift strategy adaptation without fine-tuning or additional training of a heavy LLM.

In Sotopia benchmark experiments, ALSO achieved strong overall performance improvements compared to static-based methods (nearly 84% on the Relationship dimension). It also achieved good improvements in zero-shot cross-scenario transfer experiments over the online learning baseline, demonstrating learning of transferable social dynamics rather than memorizing scenario-specific patterns.

**Compliance With Llm Reviewing Policy:**

Affirmed.

**Final Justification:**

Since I was satisfied with the manuscript and the author's rebuttal, I will keep my score (6).

**Key Questions For Authors:**

1. In the paper, the knowledge transfer was indirectly verified by the cross-scenario generalization (Figure 6). How does this actually work? **What pairs of scenarios are affecting each other's performance?** For example, if there had been analysis showing the proximity structure in the strategy embedding space or demonstrating how specific strategy updates affect the predictions of neighboring strategies, the persuasiveness of the argument would have been significantly enhanced.

2. Why is the training signal generated by DeepSeekV3.2, but the final Sotopia judgement done by a different model of GPT-4o? Do they generally provide agreeing reward? **What could be the consequence of having different evaluator models?**

3. (minor) For the score smoothing, the main manuscript (line 283) indicates that $\lambda$ = 0.7, but Table 10 in the appendix says the score decay parameter is 0.9. Is this a typo, or are these two different parameters?

**Limitations:**

I explicitly addressed most potential limitations and additional points I would have liked to include in the Strengths and Weaknesses section.

**Strengths And Weaknesses:**

# 1. Soundness

- **Strength**
   - The results consistently show the supremacy of ALSO with two different backbones.
   - Good technical integrity with well-versed appendices, such as disclosing prompts/parameters/LLM budgets, showing standard errors, and conducting additional case studies.
   - **Well-designed ablation studies for diverse dimensions**, including one-way/bilateral analysis, w/ and w/o social context, zero-shot knowledge transfer, and heterogeneous model pairing.

- **Weakness**
   - As the authors pointed out, different from EXP3, ALSO does not have theoretical boundaries for regret, and its convergence is also not guaranteed.  Including a plot of the empirical regret trajectory would be a good addition to the paper.
   - Surrogate architecture is fixed with a 2-layer MLP without ablation analysis. Since most of the 'additional' cost comes from training this surrogate, more analysis on "how simple it could be?" would be interesting (can a simple linear model achieve similar performance?)

***

# 2. Presentation

- Strength
   - The narrative is strong and easy to follow.
   - Clear visuals with well-drawn schematics (although Figure 1 has a low-dpi image in it).  In particular, **Figure 2 clearly contrasts failure and success of social settings**, underscoring the necessity of ALSO.
   - Good positioning within the current literature, exactly points out why ALSO is necessary and different from previous works (and pointing out their implicit assumptions for stationarity).

- Weakness
   - Although the paper clearly points out a variety of previous works for persona adaptation strategies (such as Sotopia-$\pi$ and Sotopia$\Omega$), **there are several other related studies that aimed to adaptively change their reasoning patterns or persona**, which I think are worth mentioning. For instance, check "Adaptive Thinking via Mode Policy Optimization for Social Language Agents" (Minzheng Wang et. al., 2024) and "Dynamic Personality in LLM Agents: A Framework for Evolutionary Modeling and Behavioral Analysis in the Prisoner’s Dilemma" (Weiqi Zeng et. al., 2025)

***

# 3. Significance

- Strength
   - At a time when the use of social simulation by LLM agents is rapidly increasing, this framework overcomes the limitations of static personas and adapts in real-time to dynamic counterparts.
   - This problem is applicable across a wide range of domains that employ LLM debate, including negotiation systems, conversational agents, and multi-agent reinforcement learning.
   - Especially, the **strategy selection trajectories in Figure 10 alone are strongly intriguing**: an additional case study solely on this (with socio-cognitive analysis) could be a good stand-alone research.

- Weakness
  - If I had to pick one, a potential weakness could be using only a single benchmark (Sotopia) and a relatively small number of samples in Sotopia-Hard, but this is not a critical problem that weakens the main message.

***

# 4. Originality

- Strength
   - **Seeing this problem as an adversarial multi-armed bandit implies a non-stationary situation**, which was implicitly assumed by many previous works in this domain. It could be a pioneering study on context-based prompt adaptation in the LLM debate.
   - Incorporating dialogue history as a social context for prompt optimization also matches well with the main assumption of this study, and was absolutely necessary.

- Weakness
  - Methodologically, all of the individual components of ALSO were previously proposed, such as EXP3, neural armed bandit, score smoothing, context embedding, etc. Surprisingly, the paper does not present ablation studies to justify these model selections.
   - Even for the combination perspective, **INSTINCT (one of the baselines, and often the best among baselines) shares a lot in common with ALSO**, using surrogate models and feeding transformer embedding to it for prompt optimization.

---

> ### Author Rebuttal · Authors · 2026-03-31
>
> We sincerely thanks for the careful and insightful review, as well as the generous recognition of our motivation, formulation, and experimental results.
> ## Response to W1.1: Regret bound; a plot of the empirical regret trajectory
>
> For standard computing the formal pseudo-regret (`Eq. 7`) requires evaluating **all K arms at every turn**, which may violates in practical and explodes LLM costs. Therefore, ALSO adopts adversarial online learning as a **design rationale for robustness** rather than claiming formal regret guarantees (`lines 204–207, p.4`).
>
> As an regret alternative, we plot the average per-turn reward trajectory for Sotopia-Hard [TrajectoryImage](https://anonymous.4open.science/r/ALSO-67D5/im1.png). The curve demonstrates rapid improvement (turn 1 to 5) and convergence thereafter.
> ## Response to W1.2: A simple linear model achieve similar performance?
>
> | Architecture | Goal | Overall |
> | :--- | :--- | :--- |
> | Linear | 7.44 | 3.65 |
> | 1-Layer | **7.77** | **3.73** |
> | 2-Layer (our) | 7.11 | 3.53 |
>
> A linear model underfits complex dialogue states and our original **2-layer MLP overfits**. Our new ablation on Sotopia-Hard reveals that a **1-layer MLP achieves the best balance**. We appreciate this insight, as it helped us identify a more optimal architecture.
> ## Response to W2: Related Works
> These works have **already been cited** in our paper. *AML `cite:line 104, p2 ref: line558 p11`* is cited noting that it "promotes diverse social reasoning patterns". *Dynamic Personality `cite:line 60, p1 ref: line598 p11`* is cited to support our premise that "relying solely on static personas often yields stereotypical and homogeneous behaviors". However, we acknowledge their connection to our methodology could be stronger. In the revision, we acknowledge their connection to our methodology deserves deeper discussion in the related work section.
>
> ## Response to W4: Lack of ablation studies justifying model selections
>
> We have conducted partial ablation studies in Section 6. We conducted comprehensive ablation studies on a subset of Sotopia, isolating the impact of each core component:
>
> | Variant | Goal | Relation | Knowledge | Overall |
> | :--- | :--- | :--- | :--- | :--- |
> | ALSO (ours) | **7.93** | **3.07** | **6.46** | **3.91** |
> | w/o EXP3 (ε-greedy) | 7.50 | 2.71 | 5.32 | 3.61 |
> | w/o Score Smoothing | 7.57 | 2.25 | 5.39 | 3.57 |
> | w/o Context Embedding | 7.43 | 2.64 | 4.82 | 3.51 |
> | w/o Neural Surrogate | 6.89 | 2.00 | 4.93 | 3.33 |
>
> Removing the Neural Surrogate causes the largest drop (−0.58 Overall); removing Score Smoothing most severely impacts Relationship (3.07→2.25); replacing EXP3 with ε-greedy degrades to 3.61, highlighting adversarial design's robustness against non-stationary dynamics.
>
> Regarding overlap with INSTINCT: while both use neural surrogates, INSTINCT applies NeuralUCB under a stationarity assumption. ALSO's adversarial weighting and score decay handle non-stationarity—confirmed by the ablations and ALSO's consistent outperformance of INSTINCT on Sotopia-Hard.
> ## Response to Q1: Cross-scenario generalization
>
> **Generalization Experiment Setup.** As see in [Setup](https://anonymous.4open.science/r/ALSO-67D5/im2.png), the surrogate is trained **sequentially** on source scenarios (Tr1–Tr7), inheriting weights across scenarios, then frozen and evaluated directly on unseen test scenarios (Te1–Te7) without further updates—analogous to zero-shot transfer or continues update surrogate.
>
> We have conducted an additional analysis examining how **reward updates propagate across semantically similar strategies**. As shown in the [Similar](https://anonymous.4open.science/r/ALSO-67D5/im3.png), the neural surrogate successfully transfers reward signals to neighboring strategies in embedding space, but the influence is **not monotonically consistent**—predictions are also modulated by dialogue context. This confirms that our surrogate captures both **semantic proximity** and **contextual relevance**, rather than performing naive interpolation.
> ## Response to Q2: Consequence of different turn evaluator
>
> We use **GPT-4o for final evaluation to ensure consistency with Sotopia benchmark requirement**. We ablate the turn-level evaluator while keeping GPT-4o as the final judge across all conditions:
>
> | Model | Goal | Relation | Knowledge | Overall |
> | :--- | :--- | :--- | :--- | :--- |
> | DeepSeek-V3.2 (Ours) | **7.93** | 3.07 | **6.46** | **3.91** |
> | 4o | 7.61 | 3.07 | 5.54 | 3.78 |
> | 4o-mini | 7.75 | **3.18** | 6.14 | 3.85 |
> | Qwen-2.5-72B | 7.04 | 2.64 | 5.86 | 3.59 |
>
> DeepSeek-V3.2 achieves the best overall score (3.91), demonstrating that ALSO is robust to evaluator choice and that DeepSeek-V3.2 serves as a high-quality and cost-efficient shaping signal that transfers well to GPT-4o's final judgments.
>
> ## Response to Q3: Inconsistency Score decay
> Thank you for catching this. It is a typo in line 283; the correct value is **0.9**, as listed in Table 10. We will correct it.

---

> > ### Author Rebuttal · Reviewer_d8mj · 2026-04-03
> >
> > Thanks for all the replies and additional experiments. In particular, while I was a bit surprised to learn that a 2-layer MLP overfits, I’m glad I managed to pinpoint an interesting aspect. Systems that can be adequately modeled by a 1-layer MLP may actually take the form of functions that can be approximated by very simple nonlinear equations (which might even be nearly analytic). If they are this simple, it should be possible to probe the MLP's input-output behavior to immediately verify the form of the reward structure that the surrogate model has actually learned.
> >
> > My other concerns (mostly minor) were all resolved, and I kept my score.

---

> > > ### Author Response · Authors · 2026-04-07
> > >
> > > We sincerely thank Reviewer for confirming that all concerns have been fully resolved. **We particularly appreciate the insightful observation** regarding the 1-layer MLP's implications—the suggestion that the learned reward structure may take a simple, nearly analytic form is a compelling direction for interpretability analysis. We plan to probe the surrogate's input-output behavior as suggested, which could yield valuable insights into the underlying reward dynamics of social simulation.
> > >
> > > **We thank the reviewer for the constructive and technically stimulating discussion throughout the review process.**

---

### Official Review · Reviewer_y2x9 · 2026-03-13

**Soundness:** 3
**Presentation:** 3
**Significance:** 3
**Originality:** 2
**Overall Recommendation:** 4
**Confidence:** 2

**Summary:**

This paper proposes ALSO, an adversarial online strategy optimization, for online strategy optimization in multi-agent social simulation. To do this, they first formulate multi-turn interactions as an adversarial bandit problem and propose to treat combinations of personas and strategy instructions as arms. To optimize the strategy, ALSO combines an exponential-weights selector with a lightweight neural surrogate that generalizes sparse feedback across strategies. Finally, extensive experiments on Sotopia show consistent improvements in social performance.

**Compliance With Llm Reviewing Policy:**

Affirmed.

**Final Justification:**

My main concern was the $K$ social strategies, which initially appeared to be handcrafted. Based on the authors’ response, I now understand that these strategies are generated through a two-stage process that combines manual design with LLM-based generation, and that they are grounded in social science theory. This clarification partly alleviates my concern, and accordingly, I have raised my score to a positive one.

**Key Questions For Authors:**

1.	The paper states that the social strategies, $\Sigma$ = {$\sigma_1, …, \sigma_K$}, are predefined. I just wonder how these social strategies are specified in practice, and according to what principles or rules they are defined?

2.	Regarding the social strategy instructions, does the choice of instruction set significantly affect performance? In other words, if the predefined set of social strategy instructions is limited, does it still improve the performance?

3.	I also have a more basic question. I can understand that Equation 7 corresponds to the regret, and a natural target objective is to minimize the regret. However, in Equation 6, why is the objective of social strategy learning formulated as minimizing two reward-related terms?

**Limitations:**

The authors do not discuss the limitations of their method. One potential limitation may lie in generalization, since the current design still appears to rely on some degree of domain knowledge. The paper has no potential negative societal impact.

**Strengths And Weaknesses:**

Strengths: The structure of the paper is clear and easy to follow. The paper solves an interesting problem in social simulation and is the first framework for online strategy optimization in multi-agent social simulation. The experiments and analysis are extensive.

Weaknesses: The method appears to rely on a certain degree of manual design. In particular, the $K$ social strategies seem to be handcrafted. Moreover, the following optimization method is built on these predefined social strategies, since the method mainly focuses on selecting the optimal instruction from this fixed strategy set.

---

> ### Author Rebuttal · Authors · 2026-03-31
>
> We thank the reviewer for recognizing **ALSO as the first online strategy optimization framework** for multi-agent social simulation, and for the positive assessment of our presentation clarity and the extensiveness and rigor of our experimental analysis.
> ## Response to Weaknesses and Q1: Predefined social strategies, and how these are specified in practice
> We acknowledge the strategy set is predefined, but clarify:
>
> **(1) ALSO supports dynamic strategy spaces.** The surrogate operates on continuous embeddings (`Eq. 11, P5`), so new strategies only require embedding—no retraining needed. Using surrogate scores to identify top-performing strategies, we prompt an LLM to generate paraphrases, dynamically expanding the pool:
>
> | Variant | Goal |  Overall |
> |---|---|---|
> | ALSO | 7.93 |  3.91 |
> | ALSO + Learnable | **8.39** | **3.92** |
>
> The gain confirms ALSO readily extends to strategy discovery. Therefore, **ALSO is not handcrafted, it fully supports autonomous strategy discovery and optimization, adaptively expanding the pool.**
>
> **(2) Strategies are grounded in social science theory, not handcrafted.** Following prior strategy injection frameworks such as Sotopia-Ω and AML. Our 12 strategies (`Table 5, P13`) derive from 6 established social science theories:
>
> | Category | Theory |
> |:---|:---|
> | Cooperative | Integrative Negotiation / BATNA [1] |
> | Competitive | Strategic Conflict & Leverage [2] |
> | Strategic | Dramaturgical Self-Presentation [3] |
> | Rational | Tit-for-Tat / Reciprocity [4] |
> | Diplomatic | GRIT Tension Reduction [5] |
> | Exploratory | Client-Centered Active Listening [6] |
>
> The strategy space is generated by prompting an LLM following DSI [7] (`Appendix B.4, P15`). Each strategy encodes principles grounded in its theoretical origin. We will **add these citations and describe the theoretical elements in the main text of the revision.**
>
> [1]: Fisher et al., *Getting to Yes*, 2011.
> [2]: Schelling, *The Strategy of Conflict*, 1960.
> [3]: Goffman, *The Presentation of Self in Everyday Life*, 1959.
> [4]: Axelrod & Hamilton, *Science*, 1981.
> [5]: Osgood, *An Alternative to War or Surrender*, 1962.
> [6]: Rogers, *Client-Centered Therapy*, 1951.
> [7]: Zhang et al., Sotopia-Ω, arXiv:2502.15538, 2025.
> ## Response to Q2: Does the choice of strategy instruction set affect performance?
>
> Strategy space diversity and size do affect performance. **However, ALSO is not limited to specific strategies — it finds the most suitable one from whatever pool is available.** Even with only 2 categories social theory, ALSO maintains comparable Overall to Vanilla (3.01 vs. 3.02), confirming no degradation under constrained spaces. With 4 categories it surpasses Vanilla by +12.6%, and 6 categories yields +29.5%.
>
> **Table 3: Strategy diversity (fixed 12 arms).**
> | #Categories | Goal |  Overall |
> |---|---|---|
> | 2 | 6.06 |  3.01 |
> | Vanilla | 6.52 |  3.02 |
> | 4 | 6.84 | 3.40 |
> | 6 | **7.93** | **3.91** |
>
> **Table 4: Strategy pool size.**
> | Pool Size | Goal | Overall |
> |---|---|---|
> | Vanilla | 6.52 |  3.02 |
> | 6 | 6.79 |  3.45 |
> | 12 (default) | **7.93** |  **3.91** |
> | 24 | 7.21 | 3.61 |
> | 48 | 6.75 |  3.29 |
>
> For the size of strategy space under Sotopia, both overly small and overly large pools lead to suboptimal trade-offs—as the arm space grows, exploration becomes difficult, making it harder to identify the optimal strategy within a limited interaction budget.
> ## Response to Q3: Eq. (6), the objective of social strategy learning
>
> Eq. (6) is a **motivational formulation rather than a tractable optimization target**—it illustrates the core difficulty of strategy learning: how to optimize a policy under non-stationary conditions given only sparse episode-level reward signals, which must supervise non-stationary turn-level expected rewards.
>
> **Directly using episode-level $r_i$ is hard due to too sparse reward singal** and introduces severe credit assignment difficulties across $L$ turns. Following Sotopia-Ω and AML, we introduce per-turn shaping rewards as a denser signal. As shown in Algorithm 1 (`line 245, p.5`), we optimize $\pi_i$ via MSE, treating per-turn realized reward as a **fixed** observed signal from the LLM evaluator. What we truly optimize is the turn-level expected reward $\mathbb{E}[r_l^{(i)} \mid s_l, a_l^{(i)}]$ to align with it—improving surrogate prediction accuracy for more reliable arm evaluation and selection. The remaining challenge is non-stationarity: as opponent policies $\pi_j$ co-evolve, the expected reward landscape continuously shifts, motivating the adversarial formulation. Therefore, Eq. (6) characterizes the optimization challenge common in social simulation—**sparse feedback and non-stationarity**—that ALSO is specifically designed to address.
>
> **We thank the reviewer for the valuable feedback and hope our responses help better evaluate our work.**

---

> > ### Author Rebuttal · Reviewer_y2x9 · 2026-04-03
> >
> > Thanks for the authors' response. I am still not fully clear about the response for Weaknesses and Q1.  Appendix B.2 provides 12 social strategies directly in Table 5. My question is how these social strategies were obtained, instead of whether ALSO supports dynamic strategy spaces. Although these strategies are grounded in social science theory, they still seem to be manually specified, right?

---

> > > ### Author Response · Authors · 2026-04-07
> > >
> > > Thank you for the clarification. We sincerely apologize that our previous response did not directly address your core question. You asked a very clear question: **How was the strategy space obtained?** Below is our explicit answer.
> > >
> > > ## Two-Stage Strategy Generation Process
> > >
> > > Our strategy generation follows a two-stage pipeline: **Theory-Grounded Categorization (Manual) → Strategy Instruction Generation (Automated via LLM)**.
> > >
> > > ### **Stage 1: Theory Category Selection (Manual)**
> > >
> > > We selected 6 theoretical categories from established social science literature as high-level semantic anchors, as mentioned in our previous rebuttal. We acknowledge this step involves manual selection. However, we emphasize that these categories are drawn from classic social science theories with decades of empirical validation, not from researchers' subjective judgment. **This approach is fully consistent with prior work in this domain (Sotopia-Ω [1], AML [2])**, which similarly leverage theory-grounded categorizations to structure their strategy spaces.
> > >
> > > ### **Stage 2: Strategy Instruction Generation (Fully Automated via LLM Paraphrasing)**
> > >
> > > Given each theoretical category, we follow a prompt optimization protocol such as APE [3], using LLMs to automatically generate concrete strategy instruction texts. For the default 12-strategy configuration, we generate 2 instructions per category. In our pool size ablation experiments (24/48 arms), we similarly expand semantic variants via LLM paraphrasing. **The specific textual content of all strategy instructions is entirely LLM-generated, not manually written.**
> > >
> > > The paraphrasing template is described in Appendix B.4 (page 15) and implemented via the following prompt template:
> > >
> > > ```
> > > You are an expert in social psychology and negotiation theory. Your task is to
> > > generate semantically equivalent paraphrases of social strategies.
> > >
> > > Original Strategy: {strategy_name}: {strategy_description}
> > > Theoretical Basis: {theory}
> > >
> > > Instructions:
> > > 1. Generate {n} paraphrased versions of this strategy.
> > > 2. Each paraphrase must:
> > >    - Preserve the core behavioral intent and theoretical grounding.
> > >    - Use different wording, sentence structures, and examples.
> > >    - Be directly usable as an agent prompt.
> > > 3. Vary the linguistic style: some formal, some conversational.
> > > 4. Do NOT change the underlying negotiation tactic.
> > > 5. Start with "In your response, "
> > >
> > > Output Format:
> > > {
> > >   "original_id": "{strategy_id}",
> > >   "paraphrases": [
> > >     {"id": "{strategy_id}_v1", "description": "..."},
> > >     {"id": "{strategy_id}_v2", "description": "..."}
> > >   ]
> > > }
> > > ```
> > >
> > > The corresponding code implementation is available at:
> > > https://anonymous.4open.science/r/ALSO-67D5/experiments/dynamic_observation/paraphrase_agent_profiles_chain.py
> > >
> > >
> > > **The current theory-grounded approach ensures interpretability and grounding in validated social science principles**, which we believe is a principled design choice for a baseline system. However, the framework's flexibility allows seamless integration of fully automated strategy generation methods without manual design, as we showed in the dynamic strategy space experiment in our previous response.
> > >
> > > We will explicitly describe this two-stage process in the revised manuscript to clarify the respective roles of manual design and automated generation.
> > >
> > > [1] Sotopia-Ω: Dynamic Strategy Injection Learning and Social Instruction Following Evaluation for Social Agents.
> > > [2] Think on Your Feet: Adaptive Thinking via Reinforcement Learning for Social Agents.
> > > [3] Large Language Models Are Human-Level Prompt Engineers.
> > >
> > > **We sincerely hope this response directly addresses your concern and would be grateful if you could re-evaluate our work in light of this clarification.**

---

### Official Review · Reviewer_k8Li · 2026-03-13

**Soundness:** 4
**Presentation:** 3
**Significance:** 3
**Originality:** 2
**Overall Recommendation:** 5
**Confidence:** 4

**Summary:**

This paper introduces ALSO (Adversarial onLine Strategy Optimization), a framework for improving the adaptability of LLM agents in multi-turn social interactions. The authors argue that many existing agents rely on fixed personas, which often leads to rigid and homogeneous behavior that fails to adapt to changing environments or strategic opponents. To address this, ALSO formulates strategy selection as an adversarial multi-armed bandit problem. Each arm corresponds to a discrete strategy instruction (e.g., integrative negotiation or reciprocity triggers) that can be appended to the agent’s base persona at each turn. To deal with sparse and non-stationary feedback in long dialogues, the framework combines a lightweight neural surrogate model that predicts rewards from interaction histories with an EXP3-style exponential-weights selector with score smoothing. Experiments on Sotopia and Sotopia-Hard show that ALSO consistently outperforms static baselines and existing prompt optimization methods, particularly in improving relational outcomes in negotiation settings.

**Compliance With Llm Reviewing Policy:**

Affirmed.

**Final Justification:**

The rebuttal addressed a meaningful portion of my concerns and changed my evaluation from 3 to 5. I was convinced by the authors’ clarification that the core contribution is the online adaptation framework, rather than strategy injection itself, and the added ablations on strategy pool size, diversity, and learnable expansion provide useful supporting evidence. I also found the discussion of baseline selection and data efficiency helpful.

Overall, I view the paper as original, technically interesting, and clearly presented, with strong significance for social agent learning. While the reliance on a discrete strategy space and the limited scaling with larger strategy pools still leave some room for improvement in soundness and scope, the rebuttal substantially strengthened my confidence in the work. For these reasons, I support acceptance.

**Key Questions For Authors:**

Q1. Following W1, How sensitive is the neural surrogate's convergence to the size and diversity of the predefined strategy space? Have you experimented with scaling the strategy pool beyond the current 12 options, and does the surrogate effectively generalize across a much larger action space?

Q2: Would the strategies be somewhat learnable skills, rather than fixed prior?

Q3: Following W2, could you add the relevant baselines for comparison?

**Limitations:**

Yes.

**Strengths And Weaknesses:**

S1. The paper has a clear and elaborated problem formulation of the problem as an adversarial multi-arm bandit problem in social interactions. The introduction of strategies in addition to the persona is an insightful prior in social environments.

S2. The paper introduces a light neural surrogate for reward prediction, which is novel and practical introduction of efficient social learning in combination to multi-arm bandit starteiges.

S3. Empirical soundness and extensiveness of analysis. The experiment results show consistent over-performance versus baselines on Sotopia and especially Sotopia-Hard.

W1. Not trying to pick on novelty, but the main contribution of this paper seems to be a complimentary work around surrogate reward method for strategic injection, given existing works on directly injecting social strategies, which limits its impact.[1]

W2. Baselines are not complete. Several other related works are relevant in terms of turn-level reward and strategy injection [2] [3]  should be cited and compared due to the high relevance.

W3. The strategies are fixed set of 12 discrete strategies and are developed as prior knowledge rather than learned skills, which simplifies the formulation but limits the paper's scope.

[1] Sotopia-Omega:  Dynamic Strategy Injection Learning and Social Instruction Following Evaluation for Social Agents https://arxiv.org/pdf/2502.15538
[2] Sotopia-RL: Reward Design for Social Intelligence https://arxiv.org/pdf/2508.03905
[3] Think on your Feet: Adaptive Thinking via Reinforcement Learning for Social Agents https://arxiv.org/html/2505.02156v2

---

> ### Author Rebuttal · Authors · 2026-03-31
>
> Thank for the positive of our problem formulation, surrogate design, and empirical analysis.
> ## Response to W1 and Q1: Directly injecting social strategies, which limits its impact
> We thank  for the thoughtful feedback. We respectfully clarify that ALSO's **core contribution is not strategy injection**, but the **online adaptive formulation — a fundamentally different paradigm from prior work**. **Existing methods** such as Sotopia-Ω [1], Sotopia-RL [2] and Think on Your Feet [3] **are inherently offline**, relying on pre-collected data or extra planners. **ALSO is the first online framework** for social simulation, formulating strategy selection as an adversarial bandit with: (1) **non-stationarity robustness** via randomized exploration; (2) **sample-efficient generalization** via a neural surrogate; (3) **natural scalability** to new strategy spaces. No offline data or retraining required.
> ## Response to Q1: Convergence to the size and diversity
> We conducted two ablation studies on a Sotopia subset.
> **① Size Ablation.** We retain 6 strategy categories and use LLM paraphrases to expand the pool to 6, 12, 24, and 48 arms.
> | Pool Size | Goal | Rel. | Know. | Overall |
> |---|---|---|---|---|
> | 6 | 6.79 | 2.25 | 5.25 | 3.45 |
> | 12 (default) | **7.93** | **3.07** | **6.46** | **3.91** |
> | 24 | 7.21 | 2.32 | 5.89 | 3.61 |
> | 48 | 6.75 | 1.85 | 5.17 | 3.29 |
>
> Performance peaks at 12 arms. The drop at 24/48 is due to **increasing exploration difficulty as the arm space grows under sotopia** with more arms to explore.
>
> **② Diversity Ablation.** We fix 12 arms and vary category prototypes (2, 4, 6), each paraphrased to fill the budget.
> | #Diversity | Goal | Rel. | Know. | Overall |
> |---|---|---|---|---|
> | 2 | 6.06 | 1.22 | 4.97 | 3.01 |
> | 4 | 6.84 | 2.24 | 5.41 | 3.40 |
> | 6 | **7.93** | **3.07** | **6.46** | **3.91** |
>
> Diversity monotonically improves performance (+29.9% Overall from 2→6).  Both ablations confirm **ALSO scales naturally, with exploration budget as the primary bottleneck rather than model capacity**.
> ## Response to W2 and Q3: Baselines are not complete
> We thank for the suggestion. Both works are already cited,   Sotopia-RL  [2] in `Line 102 P2`, and  Think on Your Feet  [3]  which corresponds to the Version 1 of "Adaptive Thinking via Mode Policy Optimization" cited as AML in `Line 104 P2`. But We will expand the Related Work to more thoroughly discuss both works.
>
> Regarding empirical comparison, we respectfully note **these methods operate under fundamentally different paradigms**:
>
> | Aspect | [2] Sotopia-RL | [3] Think on Your Feet | ALSO (Ours) |
> |:---|:---|:---|:---|
> | Paradigm | Offline SFT/RL | Offline SFT/RL | **Online learning** |
> | Training Data | Pre-collected episodes | Pre-collected episodes | **Real-time interaction** |
> | New Strategy Space | Requires retraining | Requires retraining | **Easy online adaptation via strategy embedding** |
> | Cross-model Transfer | Tied to fine-tuned LLM | Tied to fine-tuned LLM | **Model-agnostic (`Fig. 7, p8`)** |
>
> Our baselines (OPRO, EvoPrompt, INSTINCT) are selected because they operate in the same online loop with matched LLM call budgets (`Table 1 P6`), enabling fair comparison.
>
> Nevertheless, to address the reviewer's concern, we conducted a preliminary comparison under a Qwen-7B-Instruct self-play setting on a subset:
> | Method | Goal | Overall |
> |:---|:---|:---|
> | Sotopia-Ω [1] (DSI) | 7.31 | 3.51 |
> | Think on Your Feet [3] (AMPO) | 7.85 | 3.54 |
> | ALSO (Ours) | 7.26 | **3.58** |
> Despite requiring **no offline training data** — two orders of magnitude less than the ~2k episodes used by offline approaches — **ALSO achieves competitive overall performance (3.58)**, demonstrating that online adaptation constitutes a **highly effective and data-efficient paradigm**.
> ## Response to W3 and Q2: Strategies be somewhat learnable skills？
> We acknowledge the fixed strategy set but clarify that **ALSO inherently supports dynamic strategy spaces and evolution** — the surrogate predicts rewards in continuous embedding space (`Eq. 11 P5`), so any new strategy only needs embedding, no retraining required. The size ablation validates this. The fixed prior is a practical choice for experiments. With longer interactions, ALSO naturally accommodates dynamically expanding pools.
>
> The suggestion on learnable strategies is inspiring. Since **ALSO provides online reward estimates**, we use surrogate scores to select top strategies and prompt an LLM to generate paraphrases, dynamically expanding the pool. We validated this on a  subset:
> | Variant | Goal |  Overall |
> |---|---|---|
> | ALSO | 7.93 |  3.91 |
> | ALSO + Learnable | **8.39** | **3.92** |
> The gain with comparable Overall and better Goal confirms ALSO readily extends to strategy discovery and dynamic strategy space.
>
> **We sincerely thank again for the valuable and inspiring feedback. We hope our responses and additional experiments sufficiently address your concerns and help you better evaluate our work.**

---

> > ### Author Rebuttal · Reviewer_k8Li · 2026-04-06
> >
> > The rebuttal addressed a meaningful portion of my concerns. I appreciate the authors’ clarification that the main contribution is not strategy injection itself, but the online adaptation framework for strategy selection under multi-turn social interactions. The additional ablations on strategy pool size and diversity are very helpful as they support the claim in a clear way. I also appreciate the authors extensively listing some baselines and the explanation for why certain baselines are not comparable, while also denoting the data effectiveness of their method. The extra evidence from the learnable strategy space is also convincing in terms of dynamic strategy pools.
> >
> > That said, I still think the paper’s impact is somewhat constrained by the use of explicit, discrete strategy space, and the framing of the problem as multi-arm bandit, even if the rebuttal makes a reasonable case of dynamic and learnable strategy selections. Similarly, despite the monotonically increasing performance with diversity, the performance does not scale up with pool size, suggesting a limitation to the scalability of this method to larger and more granular strategy selections. However, considering the breadth of supporting evidences and the technical novelty of fully online framework, I raise my score from 3 to 5 to acknowledge the significance and soundness of this work.

---

> > > ### Author Response · Authors · 2026-04-07
> > >
> > > **We sincerely thank the reviewer for the thoughtful re-evaluation and for raising the score.** We are glad that our clarifications were found convincing and helpful, including the online adaptation framework as the core contribution, ablations on strategy pool size/diversity, baseline comparisons with data efficiency analysis, and the learnable strategy space evidence.
> > >
> > > Regarding the noted directions on continuous strategy spaces and pool size scaling, we appreciate the reviewer's forward-looking perspective. We plan to explore continuous strategy representations (e.g., soft prompts or latent strategy embeddings) and techniques to further improve the scalability of our method in future work, and will include these discussions in the revised paper.
> > >
> > > **We again thank the reviewer for the constructive engagement throughout the review process.**

---

### Decision · Program_Chairs · 2026-04-30

**Decision:**

Accept (regular)

**Comment:**

The paper presents a novel and principled approach for adapting in multi-turn dialogues.  Through solid, empirically designed experiments, the paper demonstrates the superiority of the approach over baseline methods.  One reviewer maintains a "negative" overall rating, but acknowledges strengths of the paper.  I feel the authors provided a reasonable response to the concerns expressed by the reviewer.  Hence, I recommend that the paper be accepted to ICML.